



**Drivers of diffusive lake CH₄ emissions on daily to multi-year time scales**
Joachim Jansen[1,2], Brett F. Thornton[1,2], Alicia Cortes[3], Jo Snöälv[4], Martin Wik[1,2], Sally MacIntyre[3]
and Patrick M. Crill[1,2]
[1]Department of Geological Sciences, Stockholm University, Stockholm, Sweden
[2]Bolin Centre for Climate Research, Stockholm, Sweden
[3]Marine Science Institute, University of California at Santa Barbara, Santa Barbara, USA
[4]Department of Geography, University of Exeter, Exeter, UK
Corresponding author: Joachim Jansen (joachim.jansen@geo.su.se)



**Abstract**
Lakes and reservoirs are important emitters of climate forcing trace gases. Various environmental drivers
of the flux, such as temperature and wind speed, have been identified, but their relative importance
remains poorly understood. Here we use an extensive field dataset to disentangle physical and
biogeochemical controls on the turbulence-driven diffusive flux of methane ($CH_4$) on daily to multi-year
timescales. We compare 8 years of floating chamber fluxes from three small, shallow subarctic lakes
(2010–2017, $n = 1306$) with fluxes computed using 9 years of surface water concentration measurements
(2009–2017, $n = 606$) and a small-eddy surface renewal model informed by in situ meteorological
observations. Chamber fluxes averaged $6.9 \pm 0.3$ mg m$^{-2}$ d$^{-1}$ and gas transfer velocities ($k_{600}$) from the
chamber-calibrated surface renewal model averaged $4.0 \pm 0.1$ cm h$^{-1}$. We find robust ($R^2 \geq 0.93$, $p < 0.01$)
Arrhenius-type temperature functions of the $CH_4$ flux ($E_a' = 0.90 \pm 0.14$ eV) and of the surface $CH_4$
concentration ($E_a' = 0.88 \pm 0.09$ eV). Chamber derived gas transfer velocities tracked the power-law wind
speed relation of the model ($k \propto u^{3/4}$). While the flux increased with wind speed, during storm events
($U_{10} \geq 6.5$ m s$^{-1}$) emissions were reduced by rapid water column degassing. Spectral analysis revealed
that on timescales shorter than a month emissions were driven by wind shear, but on longer timescales
variations in water temperature governed the flux, suggesting emissions were strongly coupled to
production. Our findings suggest that accurate short- and long term projections of lake $CH_4$ emissions
can be based on distinct weather- and climate controlled drivers of the flux.



## 1. Introduction

Inland waters are an important source of the radiatively active trace gas methane ($CH_4$) to the atmosphere (Bastviken et al., 2011; Cole et al., 2007). A significant portion of sediment-produced $CH_4$ reaches the atmosphere by turbulence-driven diffusion-limited gas exchange (Bastviken et al., 2004; Wik et al., 2016b) (hereafter abbreviated to 'diffusive fluxes'). Traditionally, diffusive fluxes are measured with floating chambers (Bastviken et al., 2004) but gas exchange models are increasingly used, for example to estimate annual emissions on regional scales (Holgerson and Raymond, 2016; Weyhenmeyer et al., 2015). Fluxes computed with modelled gas transfer velocities agree to a certain extent with floating chambers and the eddy covariance technique in short-term intercomparison campaigns (Bartosiewicz et al., 2015; Crill et al., 1988; Erkkilä et al., 2018). However, long-term comparisons are needed to test the validity of flux-driver relations on which models are based across a wider range of meteorological conditions, and to identify weather- and climate related controls on the flux that are appropriate for seasonal assessments. Considering the increased use of process-based approaches in regional emission estimates (DelSontro et al., 2018; Tan and Zhuang, 2015), understanding the mechanisms that drive the components of the diffusive flux is imperative to improving emission budgets.

### 1.1 Drivers of diffusive $CH_4$ emissions

Diffusive fluxes at the air-water interface can be modelled as:

$$F = k\left(C_{aq} - C_{air,eq}\right) \quad\quad [1]$$

The flux $F$ [mg m$^{-2}$ d$^{-1}$] depends on the concentration difference across a thin layer immediately below the air-water interface ($\Delta[CH_4]$ in mg m$^{-3}$), which upper boundary is in equilibrium with the atmosphere ($C_{air,eq}$) and base represents the bulk liquid ($C_{aq}$), and is limited by the gas transfer velocity $k$ [m d$^{-1}$] (Wanninkhof, 1992). $k$ has been conceptualized as characterizing transfer across the diffusive boundary layer, although other models envision a surface renewal approach in which parcels of water intermittently are in contact with the atmosphere and $k$ depends on the frequency of these renewal events (Csanady, 2001; Lamont and Scott, 1970).

The gas transfer coefficient depends on turbulence caused by wind shear and convection and on the molecular diffusivity of the dissolved gas (see MacIntyre et al., 1995 for an overview of the thermodynamic and kinetic drivers of $k$). In a stratified water column the force of buoyancy counteracts that of wind shear, and gases may accumulate below a shallow upper mixing layer (MacIntyre et al., 2010). Conversely, thermal convection as a result of surface cooling can deepen the mixed layer and transfer stored gas to the surface (Crill et al., 1988; Eugster et al., 2003), and enhance emissions at night when the surface cools despite low wind speeds (Heiskanen et al., 2014; Podgrajsek et al., 2014b; Poindexter et al., 2016). While progress has been made in understanding how the components of $k$ vary as a function of turbulence (Tedford et al., 2014) and other factors, such as lake morphology and distance to the shoreline (Read et al., 2012; Schilder et al., 2013; Vachon and Prairie, 2013), the temporal variability and drivers of $\Delta[CH_4]$ remain poorly resolved (Loken et al., 2019; Natchimuthu et al., 2016).

$CH_4$ emissions to the atmosphere also depend on the rates of methane metabolism regulated by substrate availability and temperature-dependent shifts in enzyme activity and microbial community



structure (Borrel et al., 2011; McCalley et al., 2014; Tveit et al., 2015). Arrhenius-type relationships of
CH$_4$ fluxes have emerged from field studies (DelSontro et al., 2018; Natchimuthu et al., 2016; Wik et al.,
2014) and across latitudes and aquatic ecosystem types in synthesis reports (Rasilo et al., 2015; Yvon-
Durocher et al., 2014). However, the temperature sensitivity is modulated by biogeochemical factors
that differ between lake ecosystems, such as nutrient content (Davidson et al., 2018; Sepulveda-Jauregui
et al., 2015), methanotrophic activity (Duc et al., 2010; Lofton et al., 2014), predominant emission
pathway (DelSontro et al., 2016; Jansen et al., 2019) and warming history (Yvon-Durocher et al., 2017). In
lakes, the air-water concentration difference driving the flux (Eq. 1) is further impacted by abiotic factors
that dissociate production from emission rates, such as hydrologic inputs of terrestrially produced CH$_4$
(Miettinen et al., 2015; Murase et al., 2003; Paytan et al., 2015), redistribution of dissolved gas in the
water column (DelSontro et al., 2017; Hofmann, 2013) and storage-and-release cycles associated with
transient stratification (Czikowsky et al., 2018; Jammet et al., 2017; Vachon et al., 2019). From these
interacting functional dependencies emerge complex responses of the flux to biotic and abiotic factors.
Disentangling the physical and biogeochemical drivers of the diffusive CH$_4$ flux remains a challenge. They
respond differently to slow and fast changes in meteorological covariates (Baldocchi et al., 2001;
Koebsch et al., 2015) such that different mechanisms may explain the diel and seasonal variability of the
flux. For example, temperature affects emissions through convective mixing on short timescales and
through the rate of sediment methanogenesis on longer timescales; the diurnal cycle of insolation may
have a limited effect on production because the heat capacity of the water buffers the temperature
signal (Fang and Stefan, 1996). Similar phase lags and amplifications may lead to hysteretic flux patterns,
such as cold season emission peaks due to hypolimnetic storage in dimictic lakes (Encinas Fernández et
al., 2014; López Bellido et al., 2009) or thermal inertia of lake sediments (Zimov et al., 1997). Spectral
analysis of the flux and its components can improve our understanding of the flux variability by
quantifying how much power is associated with key periodicities (Baldocchi et al., 2001).
Here we present a high-resolution, long-term dataset (2010–2017) of turbulence-driven diffusion-limited
CH$_4$ fluxes from three subarctic lakes estimated with floating chambers ($n$ = 1306) and a gas exchange
model informed by in situ meteorological observations and surface water concentrations ($n$ = 535). We
use a surface renewal model and Arrhenius relationships of $\Delta[CH_4]$ to disentangle the abiotic and biotic
effects of temperature on the flux. We then use stochastic tools to estimate the importance of these and
other flux controls on different timescales.



## 2. Materials and Methods

### 2.1 Field site

We monitored $CH_4$ emissions from three subarctic lakes of post-glacial origin (Kokfelt et al., 2010), located on the Stordalen Mire in northern Sweden (68°21' N, 19°02' E, Fig. 1), a peatland underlain by discontinuous permafrost (Malmer et al., 2005). The mire (350 m a.s.l.) is part of a catchment that connects Mt. Vuoskoåiveh (920 m a.s.l.) in the south to Lake Torneträsk (341 m a.s.l.) in the north (Lundin et al., 2016; Olefeldt and Roulet, 2012). Villasjön is the largest and shallowest of the lakes (0.17 $km^2$, 1.3 m max. depth) and drains through water-logged fens into a stream feeding Mellersta Harrsjön and Inre Harrsjön, which are 0.011 and 0.022 $km^2$ in size and have maximum depths of 6.7 m and 5.2 m, respectively (Wik et al., 2011). The lakes are normally ice-free from the beginning of May through the end of October. Manual observations were generally conducted between mid-June and the end of September. Diffusion accounts for 17%, 52% and 34% of the ice-free $CH_4$ flux in Villasjön, Inre and Mellersta Harrsjön, respectively, with the remainder being emitted via ebullition (2010–2017; Jansen et al., 2019).

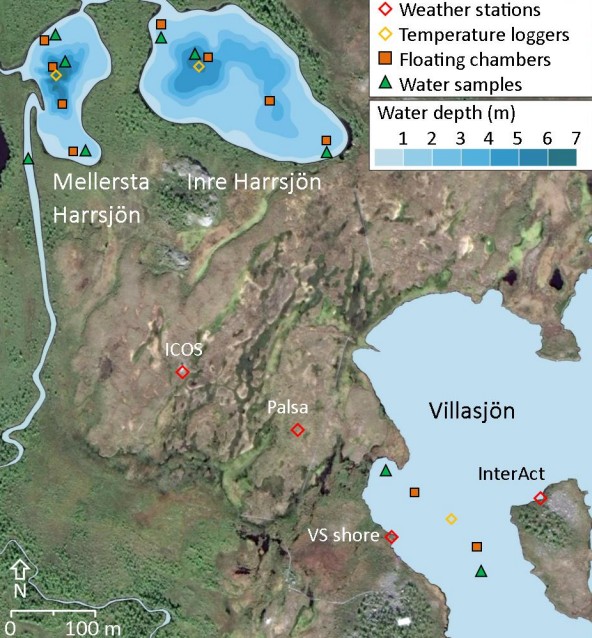

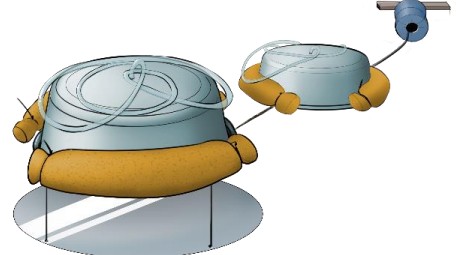

**Figure 1** – Map of the Stordalen Mire field site (left). Chamber and sampling locations are shown as they were in 2015–2017. A schematic of the floating chamber pairs is shown to the right. Lake bathymetry from Wik et al. (2011). Satellite imagery: ©Google, DigitalGlobe, 2017.



**2.2 Floating chambers**

We used floating chambers to directly measure the turbulence-driven diffusive $CH_4$ flux across the air-
water interface (Fig. 1). They consisted of plastic tubs covered with aluminium tape to reflect incoming
radiation and were equipped with polyurethane floaters and flexible sampling tubes capped at one end
with 3-way stopcocks (Bastviken et al., 2004). Depending on flotation depth each chamber covered an
area between 610 and 660 $cm^2$ and contained a headspace of 4 to 5 litres. A plastic shield was mounted
underneath one chamber of each pair to deflect methane bubbles rising from the sediment. Every 1–2
weeks during the ice-free seasons of 2010 to 2017 2–4 chamber pairs were deployed in Villasjön and 4–7
chamber pairs in Inre and Mellersta Harrsjön in different depth zones (Fig. 1). The number of chambers
and deployment intervals exceeded the minimum needed to resolve the spatiotemporal variability of the
flux (Wik et al., 2016a). Over a 24 hour period 2–4 60 mL headspace samples were collected from each
chamber using polypropylene syringes and the flotation depth and air temperature were noted in order
to calculate the headspace volume. The 24-hour deployment period was chosen to compute fluxes over
timescales which integrate diel variations in the gas transfer velocity (Bastviken et al., 2004).
The fluxes reported here are from the shielded chambers only. To check that the shields were not
reducing fluxes from turbulent processes such as convection, we compared fluxes from shielded and
unshielded chambers on days when the lake mean bubble flux was <1% of the lake mean diffusive flux
(bubble traps, 2009–2017; Jansen et al., 2019; Wik et al., 2013). Averaged over the three lakes, the
difference was statistically significant ($F_{ch,unsh} - F_{ch,sh}$ = 0.20 ± 0.16 mg m$^{-2}$ d$^{-1}$ ($n$ = 58) (mean ± 95% CI)) but
only a 6% difference from mean fluxes. Conversely, some types of floating chambers can enhance gas
transfer by creating artificial turbulence when dragging through the water (Matthews et al., 2003;
Vachon et al., 2010; Wang et al., 2015). The effect appears to be negligible for chambers of the design,
size and flotation depth used in this study (acoustic Doppler velocimeter measurements, Ribas-Ribas et
al., 2018).
**2.3 Water samples**

Surface water samples were collected at a depth of 0.2–0.4 m at 2–3 different locations in each lake (Fig.
1), at one to two-week intervals from June to October. Samples were collected from shore with a 4 m
Tygon tube attached to a floater to avoid disturbing the sediments (2009–2014) and from a rowing boat
over the deepest points of Inre and Mellersta Harrsjön (2010–2017) and at shallows (<1 m water depth)
on either end of the lakes (2015–2017) using a 1.2 m Tygon tube. In addition, water samples were
collected at the deepest point of Inre and Mellersta Harrsjön at 1 m intervals down to 0.1 m from the
sediment surface with a 7.5 m fluorinated ethylene propylene (FEP) tube. 60 mL polypropylene syringes
were rinsed thrice with sample water before duplicate bubble-free samples were collected, and were
capped with airtight 3-way stopcocks. 30 mL samples were equilibrated with 30 mL headspace and
shaken vigorously by hand for 2 minutes (2009–2014) or on a mechanical shaker at 300 rpm for 10
minutes (2015–2017). Prior to 2015, lab air – with a predetermined $CH_4$ content – was used as
headspace. From 2015 on we used an $N_2$ 5.0 headspace (Air Liquide). Water sample conductivity was
measured over the ice-free season of 2017 ($n$ = 323) (S230, Mettler-Toledo).





### 2.4 Concentration measurements

Gas samples were analysed within 24 hours after collection at the Abisko Scientific Research Station, 10
km from the Stordalen Mire. Sample $CH_4$ contents were measured on a Shimadzu GC-2014 gas
chromatograph which was equipped with a flame ionization detector (GC-FID) and a 2.0 m long, 3 mm ID
stainless steel column packed with 80/100 mesh HayeSep Q and used $N_2$ >5.0 as a carrier gas (Air
Liquide). For calibration we used standards of 2.059 ppm $CH_4$ in $N_2$ (Air Liquide). 10 standard
measurements were made before and after each run. After removing the highest and lowest values,
standard deviations of the standard runs were generally less than 0.25%.

### 2.5 Water temperature and pressure loggers

Water temperature was measured every 15 minutes from 2009 to 2018 with temperature loggers (HOBO
Water Temp Pro v2, Onset Computer) in Villasjön and at the deepest locations within Inre and Mellersta
Harrsjön. Sensors monitored the surface water in all lakes at 0.1, 0.3, 0.5, 1.0 m depth, and further at
3.0, 5.0 m (IH and MH) and at 6.7 m (MH) at the deep points. Sensors were intercalibrated prior to
deployment in a well-mixed water tank, and by comparing readouts just before ice-on when the water
column was isothermal. In this way a precision of <0.05 °C was achieved. The bottom sensors were
buried in the surface sediment and were excluded from in situ intercalibration. Water pressure was
measured in Mellersta Harrsjön (5.5 m) with a HOBO U20 Water Level logger (Onset Computer).

### 2.6 Meteorology

Meteorological data was collected from four different masts on the Mire (Fig. 1), and collectively covered
a period from June 2009 to October 2017 with half-hourly measurements of wind speed, air
temperature, relative humidity, air pressure and irradiance (Table 1). Wind speed was measured with 3D
sonic anemometers at the Palsa tower ($z$ = 2.0 m), the Villasjön shore tower ($z$ = 2.9 m), at the InterAct
Lake tower ($z$ = 2.0 m) and at the Integrated Carbon Observation System (ICOS) site ($z$ = 4.0 m). Air
temperature and relative humidity were measured at the Palsa tower, at the Villasjön shore tower
(Rotronic MP100a (2012–2015) / Vaisala HMP155 (2015–2017)) and at the InterAct lake tower. Incoming
and outgoing shortwave and long wave radiation were monitored with net radiometers at the Palsa
tower (Kipp & Zonen CNR1) and at the InterAct lake tower (Kipp & Zonen CNR4). Precipitation data was
collected with a WeatherHawk 500 at the ICOS site. Overlapping measurements were cross-validated
and averaged to form a single timeseries.

**Table 1** – Location and instrumentation of meteorological observations on the Stordalen mire, 2009–2018.

| Identifier | Period | Location | Wind | Air temp. and humidity | Radiation | Ref. |
|---|---|---|---|---|---|---|
| Palsa tower | 2009-11 | 68°21'19.68"N 19° 2'52.44"E | C-SAT 3 *Campbell Scientific* | HMP-45C *Campbell Scientific* | CNR-1 *Kipp & Zonen* | Olefeldt et al., 2012 |
| Villasjön shore tower | 2012-18 | 68°21'14.58"N 19° 3'1.07"E | R3-50 *Gill* | MP100a, *Rotronic* HMP155, *Vaisala* | REBS Q7.1 *Campbell Sci.* | Jammet et al., 2015 |
| InterAct Lake tower | 2012-18 | 68°21'16.22"N 19° 3'14.98"E | uSonic 3 Scientific *Metek* | CS215 *Campbell Scientific* | CNR-4 *Kipp & Zonen* | x |
| ICOS site | 2013-18 | 68°21'20.59"N 19° 2'42.08"E | | Weatherhawk 500 *Campbell Scientific* | | x |




**2.7 Computation of CH$_4$ storage and residence time**

The amount of stored CH$_4$ (g CH$_4$ m$^{-2}$) was computed by weighting and then adding each concentration measurement by the volume of the 1 m depth interval within which it was collected. For the upper 2 m of the two deeper lakes we separately computed storage in the vegetated littoral zone from near-shore concentration measurements, as these values could be different from those further from shore due to outgassing and oxidation during transport (DelSontro et al., 2017). We computed the average residence time of a CH$_4$ molecule by dividing the amount stored by the lake mean surface flux. Residence times computed with this approach should be considered upper limits, because we implicitly assumed that removal processes other than surface emissions, such as microbial oxidation, were negligible or took place at the sediment-water interface with minimal impact on water column CH$_4$.

**2.8 Flux calculations**

In order to calculate the chamber flux with Eq. 1 we estimated $k_{ch}$ from the time-dependent equilibrium chamber headspace concentration $C_{h,eq}(t)$ [mg m$^{-3}$] (Bastviken et al., 2004):

$$[C_{aq} - C_{h,eq}(t)] = [C_{aq} - C_{h,eq}(t_0)]e^{-\frac{K_H RTA}{V}k_{ch}t} \qquad [2]$$

where $K_H$ is Henry's law constant for CH$_4$ [mg m$^{-3}$ Pa$^{-1}$] (Wiesenburg and Guinasso, 1979), $R$ is the universal gas constant [m$^3$ Pa mg$^{-1}$ K$^{-1}$], $T$ is the surface water temperature [K] and $V$ and $A$ are the chamber volume [m$^3$] and area [m$^2$], respectively. This method accounts for gas accumulation in the chamber headspace, which reduces the concentration gradient and limits the flux (Eq. 1) (Fig. 2). For a subset of chamber measurements where simultaneous water concentration measurements were unavailable we computed the flux from the headspace concentrations alone:

$$F = c_1 M \frac{\partial x_h}{\partial t} \frac{PV}{RTA} \qquad [3]$$

where $\partial x_h/\partial t$ is the headspace mole fraction change [$10^{-6}$ ppm d$^{-1}$], $M$ is the molar mass of CH$_4$ (0.016 mg mol$^{-1}$), $P$ is the air pressure [Pa], $T$ is the air temperature [K]. Scalar $c_1$ corrects for accumulation of CH$_4$ gas in the chamber headspace and increases over the deployment time. Comparing both chamber flux calculation methods we find $c_1$ = 1.21 for 24 hour deployments (OLS, R$^2$ = 0.85, $n$ = 357). Chambers were sampled up to 4 times during deployment (at 10 minutes, 1–5 hours and 24 hours) which allowed us to compute fluxes at different time intervals.

Figure 2 illustrates the importance of the headspace correction. The headspace-corrected flux (dashed red line) equals the initial slope of Eq. 2 (solid red line) and is about 21% higher than the non-corrected flux (lower dashed black line). However, both Eq. 2 (solid red line) and Eq. 3 with $c_1$ = 1 (dashed black lines) fit the concentration data (R$^2$ ≥ 0.98 for 94% of 24-hour flux measurements). This is partly because the fluxes were low enough to keep headspace concentrations well below equilibrium with the water column, and because on average, the gas transfer velocity deviated ≤10% from its mean value over its diel cycle (Fig. 7d). Short-term measurements (upper dashed black line) may omit the need for headspace correction but can significantly overestimate the flux if – as in our study – initial chamber deployment takes place during daytime and $k$ or $\Delta$[CH$_4$] follow a diurnal pattern (Bastviken et al., 2004).



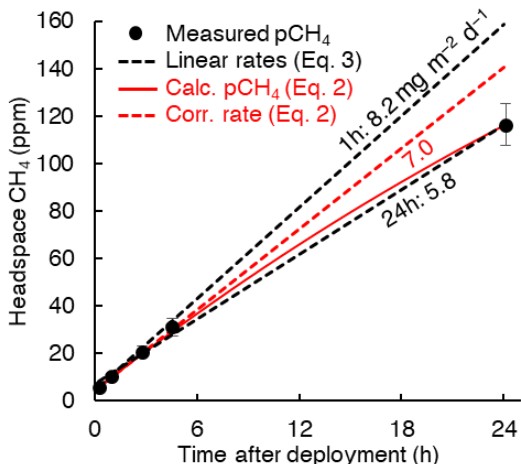

**Figure 2** – Example of chamber headspace $CH_4$ concentrations versus deployment time. Measured
concentrations (dots) are averages from 2015–2017 (0.1h) and 2011 (1h–24h); error bars represent the
95% confidence intervals. Linear regressions (dashed black lines) show the rate increase over 1 hour (two
measurements) and over 24 hours (five measurements). The solid red line represents chamber
concentrations computed with Eq. 2 using multi-year mean values of $\Delta[CH_4]$ and $k_{ch}$ (uncorrected for
headspace accumulation). The rate increase associated with the mean 24h flux corrected for headspace
accumulation is shown as a dashed red line (Eq. 1 with $k_{ch}$ from Eq. 2, or Eq. 3 with $c_1 = 1.21$). Labels
denote fluxes calculated from the linear regression slopes (Eq. 3, black) and from Eq. 2 (red).

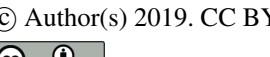



**2.9 Computing gas transfer velocities with the surface renewal model**
We used the surface renewal model (Lamont and Scott, 1970) formulated for small eddies at Reynolds
numbers >500 (MacIntyre et al., 1995; Theofanous et al., 1976) to estimate $k$:

$$k_{mod} = \alpha(\varepsilon v)^{\frac{1}{4}} Sc^{-\frac{1}{2}} \qquad [4]$$

where the hydrodynamic and thermodynamic forces driving gas transfer are expressed, respectively, as
the dissipation of turbulent kinetic energy (TKE), $\varepsilon$ [m²s⁻³], and the dimensionless Schmidt number $Sc$,
defined as the ratio of the kinematic viscosity $v$ [m²s⁻¹] to the free solution diffusion coefficient $D_0$ [m²s⁻¹]
(Jähne et al., 1987; Wanninkhof, 2014). The scaling parameter $\alpha$ has a theoretical value of 0.37 (Katul et
al., 2018), but is often estimated empirically ($\alpha'$) to calibrate the model (e.g. Wang et al., 2015). To allow
for a qualitative comparison between model and chamber fluxes we regressed $k_{ch}$ (floating chambers)
onto $(\varepsilon v)^{\frac{1}{4}} Sc^{-\frac{1}{2}}$ (surface renewal model, half-hourly values of $k_{mod}$ averaged over each chamber
deployment period), and determined $\alpha'$ = 0.24 ± 0.04 (mean ± 95% CI, $n$ = 334) (Fig. 3). When comparing
$k$-values we normalized to a Schmidt number of 600 (CO₂ at 20 °C) (Wanninkhof, 1992): $k_{600} =$
$(600/Sc)^{-0.5}k$. To enable comparison with published wind-$k$ relations we calculated the wind speed at
10 m ($U_{10}$) from the anemometer datasets following Smith (1988), assuming a neutral atmosphere.

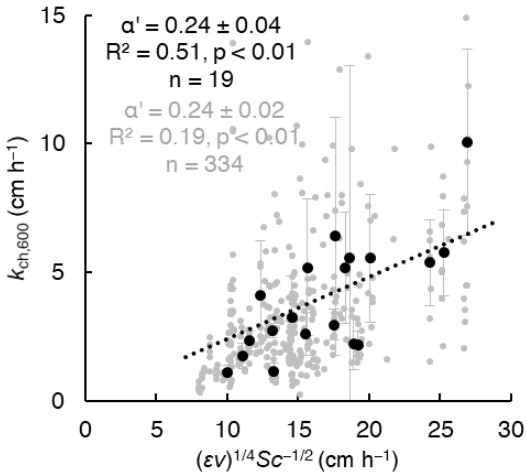

**Figure 3** – Comparison between gas transfer velocities from floating chambers (Eq. 2) and the surface
renewal model (Eq. 4 with $\alpha'$ = 1 and $Sc$ = 600, half-hourly values averaged over each chamber
deployment period). Grey dots are individual chamber deployments and black dots represent multi-
chamber means for each weekly deployment in 2016 and 2017, when concentration measurements
were taken simultaneously with, and in close proximity to the chamber measurements. Intercepts of the
linear regressions (lines) were fixed at 0. Error bars represent 95% confidence intervals of the means.

We used a parametrization by Tedford et al. (2014) based on Monin-Obukhov similarity theory to
estimate the TKE dissipation rate at half-hourly time intervals:

$$\varepsilon = \begin{cases} 0.56\, u_{*w}^3/\kappa z + 0.77\beta & \text{if} \quad \beta > 0 \\ 0.6\, u_{*w}^3/\kappa z & \text{if} \quad \beta \leq 0 \end{cases} \qquad [5]$$



where $u_{*w}$ is the water friction velocity [m s$^{-1}$], $\kappa$ is the von Kármán constant, $z$ is the depth below the
water surface (here set to 0.15 m, the depth for which Eq. 5 was calibrated). We determined $u_{*w}$ from
the air friction velocity $u_{*a}$ assuming equal shear stresses ($\tau$) on either side of the air-water interface;
$\tau = \rho_a u_{*a}^2 = \rho_w u_{*w}^2$ (MacIntyre and Melack, 1995), and taking into account atmospheric stability
(Imberger, 1985; MacIntyre et al., 2014; Tedford et al., 2014). $\beta$ is the buoyancy flux [m$^2$ s$^{-3}$], which
accounts for turbulence generated by convective mixing (Imberger, 1985):

$$\beta = \frac{\alpha_T g Q_{eff}}{c_{pw} \rho_w} \qquad [6]$$

where $\alpha_T$ is the thermal expansion coefficient [m$^3$ K$^{-1}$] (Kell, 1975), $g$ is the standard gravity [m s$^{-2}$], $c_{pw}$
[J kg$^{-1}$ K$^{-1}$] is the water specific heat and $\rho_w$ [kg m$^{-3}$] is the water density, calculated from the water
temperature and corrected for dissolved solids using conductivity measurements and a conversion factor
of 0.57 g kg$^{-1}$ / mS cm$^{-1}$. $Q_{eff}$ [W m$^{-2}$] represents the net heat flux into the surface mixed layer and is the
sum of net shortwave and long-wave radiation and sensible and latent heat fluxes. We used Beer's Law
to compute penetration of radiation into the water column across seven wavelength bands (Jellison and
Melack, 1993). Attenuation of the visible portion of the spectrum was computed from the Secchi depth
(Karlsson et al., 2010; Wik et al., 2018) with the inverse relationship from Idso and Gilbert (1974). We
further computed outgoing component of the net longwave radiation ($LW_{net}$) using the Stefan-Boltzmann
law: $LW_{out} = \sigma T^4$, where σ is the Stefan-Boltzmann constant (5.67 × 10$^{-8}$ W m$^{-2}$ K$^{-4}$) and $T$ is the surface
water temperature in K. For periods where we did not have longwave radiation data we assumed $LW_{net}$ =
−50 W m$^{-2}$. Sensible and latent heat fluxes were computed with bulk aerodynamic formula described in
MacIntyre et al. (2002). Both $Q_{eff}$ and $\beta$ are here defined as positive when the heat flux is directed out
of the water, for example when the surface water cools.

Direct measurements of turbulent dissipation rates in a small Arctic lake (1 m depth, 0.005 km$^2$) show
that Equation 5 well characterizes near-surface turbulence in small, sheltered water bodies similar to the
lakes studied here (MacIntyre et al., 2018). Eq. 5 underestimates the dissipation-suppressing effects of
stratification of the upper water column at buoyancy frequencies ($N = \sqrt{g/\rho_w \times d\rho_w/dz}$) exceeding 25
cycles per hour (MacIntyre et al., 2018). However, in the current dataset such periods of strong
stratification ($N > 25$ cph) were observed <3% of the time.

**2.10 Calculation of binned means**
We binned data to assess correlations between the flux and environmental covariates. Half-hourly values
of water temperature and wind speed were averaged over the deployment period of each chamber
(fluxes) and over 24 hours prior to the collection of each water sample (concentrations). The 24 hour
averaging period was chosen based on the mean residence time of a CH$_4$ molecule in the lake water
column. Parameters of interest (fluxes, concentrations and $k$) were then binned in 10 day, 1 °C and 0.5 m
s$^{-1}$ bins to obtain relationships with time, water temperature and wind speed, respectively. For this
calculation, lake-specific variables such as water temperature were normalized by lake to obtain a single
timeseries (divided by the lake mean, multiplied by the overall mean).

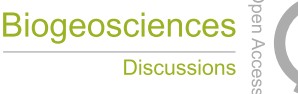

**2.11 Calculation of the empirical activation energy**

Chamber and modelled fluxes as well as surface concentrations were fitted to an Arrhenius-type temperature function (e.g. Wik et al., 2014; Yvon-Durocher et al., 2014):

$$F = e^{-E_a'/k_B T + b} \qquad [7]$$

where $k_B$ is the Boltzmann constant ($8.62 \times 10^{-5}$ eV K$^{-1}$) and $T$ is the water temperature in K. The empirical activation energy ($E_a'$, in electron volts (eV), 1 eV = 96 kJ mol$^{-1}$) was computed with a linear regression of natural logarithm of the fluxes and concentrations onto the inverse temperature (1/K), of which $b$ is the intercept.

**2.12 Timescale analysis: power spectra and climacogram**

We computed power spectra for near-continuous timeseries of the water- and air temperature and the wind speed according to Welch's method (pwelch in MATLAB 2018a), which splits the signal into overlapping sections and applies a cosine tapering window to each section (Hamming, 1989). Data gaps were filled by linear interpolation. We removed the linear trend from original timeseries to reduce red noise, and block-averaged spectra (8 segments with 50% overlap) to suppress aliasing at higher frequencies. We normalized the spectra by multiplying by the natural frequency and dividing by the variance of the original timeseries (Baldocchi et al., 2001).

We evaluated discontinuous timeseries with a climacogram, an intuitive way to visualize a continuum of variability (Dimitriadis and Koutsoyiannis, 2015). It displays the change of the standard deviation ($\sigma$) with averaging timescale ($t_{avg}$) in double-logarithmic space. Variables of interest were normalized by lake to create a single time-series at half-hourly resolution (i.e. 48 entries for each 24-hour chamber flux). To compute each standard deviation ($\sigma(t_{avg})$) data was binned according to averaging timescale, which ranged from 30 minutes to 1 year. Because of the discontinuous nature of the datasets, $n$ bins were distributed randomly across the time series. We chose $n$ = 100000 to ensure that the 95% confidence interval of the standard deviation at the smallest bin size was less than 1% of the value of $\sigma$ (Sheskin, 2007). To allow for comparison between variables we normalized each $\sigma$-series by its smallest-bin value: $\sigma_{norm} = \sigma/\sigma_{init}$. For timescales < 1 week we used 1-hour chamber observations. We use the climacogram specifically to test whether the variability of the diffusive CH$_4$ flux is enveloped by hydrometeorological variability, as for terrestrial ecosystem processes (Pappas et al., 2017).

**2.13 Statistics**

We used Analysis of Variance (ANOVA) and the t-test to compare means of different groups. The use of means, rather than medians was necessary because annual emissions can be determined by rare, high-magnitude emission events. Parametric tests were justified because of the large number of samples in each analysis, in accordance with the central limit theorem. Linear regressions were performed with the ordinary least squares method (OLS): reported $p$-values refer to the significance of the regression slope. Non-linear regressions were optimized with the Levenberg-Marquardt algorithm for non-linear least squares with confidence intervals based on bootstrap replicates ($n$ = 1999). Computations were done in MATLAB 2018a and in PAST v3.25 (Paleontological Statistics software package) (Hammer et al., 2001).



## 3. Results

### 3.1 Measurements and models

Chamber fluxes averaged 6.9 mg m$^{-2}$ d$^{-1}$ (range 0.2–32.2, $n$ = 1306) and closely tracked the temporal evolution of the surface water concentrations (mean 11.9 mg m$^{-3}$, range 0.3–120.8, $n$ = 606), with the higher values in each lake measured in the warmest months (July and August, Fig 4a,e). As expected, diffusive fluxes increased with wind speed and water temperature (Fig 4b,c). Reduced emissions were measured in the shoulder months (June and September) and were associated with lower water temperatures. We also observed abrupt reductions of the flux at wind speeds lower than 2 m s$^{-1}$ and higher than 6.5 m s$^{-1}$. Surface water concentrations generally increased with temperature and peaked in the summer months, but unlike the chamber fluxes they decreased with increasing wind speed (Fig. 4f,g). Relationships with wind speed were approximately linear, while relationships with temperature fitted an Arrhenius-type exponential function (Eq. 7). Activation energies were not significantly different between the surface water and sediment temperature ($E_a'$ = 0.90 ± 0.14 eV, R$^2$ = 0.93, $E_a'$ = 1.00 ± 0.17, R$^2$ = 0.93, respectively, mean ± 95% CI). The fluxes, concentrations, and the wind speed were non-normally distributed (Fig. 4d,h,o). Surface water temperatures (0.1–0.5 m) were normally distributed for each individual month of the ice-free season (Fig. 4n), but the composite distribution was bimodal.

Fluxes computed with the surface renewal model (Eq. 1 using $k_{mod}$) closely resembled the chamber fluxes (Eq. 3) in terms of temporal evolution (Fig. 4a) and correlation with environmental drivers (Fig. 4b,c). Despite the model's calibration with a subset of the chamber data, model fluxes were higher than the chamber fluxes in all lakes (Table 2). Model fluxes were significantly different between littoral and pelagic zones in Inre and Mellersta Harrsjön (paired t-tests, $p$ ≤ 0.02), reflecting spatial differences in the surface water concentration (Table 2). Similar to the chamber fluxes, the air-water concentration difference (Δ[CH$_4$]) explained most of the temporal variability of the modelled emissions; both $k_{mod}$ (Eq. 4) and $k_{ch}$ (Eq. 2) were functions of $U_{10}$ (Fig. 4k) and did not display a distinctive seasonal pattern (Fig. 4i). Modelled fluxes were lower at higher wind speeds and displayed a cut-off at $U_{10}$ ≥ 6.5 m s$^{-1}$, similar to the chamber fluxes, but not at $U_{10}$ < 2.0 m s$^{-1}$. The temperature sensitivity of the modelled fluxes ($E_a'$ = 0.97 ± 0.12 eV, mean ± 95% CI, R$^2$ = 0.94) did not differ significantly from that of the chamber fluxes.

**Table 2** – CH$_4$ fluxes from floating chambers and the surface renewal model, and surface CH$_4$ concentrations. 2014 was excluded from the model flux means because of a substantial bias in the timing of sample collection.

| Location | Chamber flux (mg m$^{-2}$ d$^{-1}$) | | Modelled flux (mg m$^{-2}$ d$^{-1}$) | | Surface concentration (mg m$^{-3}$) | |
|---|---|---|---|---|---|---|
| | mean ± 95% CI | $n$ | mean ± 95% CI | $n$ | mean ± 95% CI | $n$ |
| Overall | 6.9 ± 0.3 | 1306 | 7.6 ± 0.5 | 501 | 11.9 ± 0.9 | 606 |
| Villasjön | 5.2 ± 0.5 | 249 | 5.3 ± 0.7 | 149 | 8.3 ± 1.1 | 183 |
| Inre Harrsjön | 6.6 ± 0.4 | 532 | 6.9 ± 0.6 | 176 | 10.2 ± 1.0 | 211 |
| Shallow (<2 m) | 6.0 ± 0.6 | 219 | 7.6 ± 0.8 | 113 | 11.1 ± 1.3 | 133 |
| Intermediate (2-4 m) | 7.1 ± 0.6 | 212 | | | | |
| Deep (>4 m) | 6.6 ± 0.8 | 101 | 6.4 ± 0.9 | 63 | 8.6 ± 1.4 | 78 |
| Mellersta Harrsjön | 8.0 ± 0.4 | 525 | 10.4 ± 0.9 | 176 | 16.7 ± 2.0 | 212 |
| Shallow (<2 m) | 8.1 ± 0.6 | 272 | 11.1 ± 1.3 | 113 | 18.2 ± 2.7 | 134 |
| Intermediate (2-4 m) | 7.8 ± 0.7 | 154 | | | | |
| Deep (>4 m) | 8.0 ± 1.0 | 99 | 9.1 ± 1.2 | 63 | 14.1 ± 2.7 | 78 |





**Figure 4** – Scatterplots of the CH₄ flux (**a-c**), CH₄ air-water concentration difference (**e-g**) and gas transfer
velocity (**i-k**) versus time, surface water temperature and wind speed, as well as the histograms of the
aforementioned variables. In each scatter plot binned means are represented by large symbols with
error bars signifying 95% confidence intervals. Bin sizes were 10 days, 1 °C and 0.5 m s⁻¹ for time, surface
water temperature and U₁₀, respectively. Small green, blue and red dots represent individual
measurements in Villasjön, Inre Harrsjön and Mellersta Harrsjön, respectively. Open rhombus symbols in
panels **i-k** represent the buoyancy component of the gas transfer velocity, closed rhombus symbols
include both the wind-driven and buoyancy-driven components. Dashed lines in panels **b** and **f** represent
fitted Arrhenius functions (Eq. 7). Histograms of modelled (light blue) and measured (light orange)
quantities (**d,h,l**) overlap. Histograms of the surface water temperature (**m**) and U₁₀ (**o**) are stacked by
month, from June (darkest shade) to October (lightest shade).





### 3.2 Meteorology and mixing regime

The water column of all three lakes was weakly stratified throughout the ice-free season, and the mean diel mixing depths (d$\rho$/dz < 0.03 kg m$^{-3}$ m$^{-1}$ (Rueda et al., 2007)) exceeded the lake mean depths (Table 3). Figure 5 shows a timeseries of the mixing depth and water temperature in the deeper lakes, along with wind speed, air temperature and precipitation for the ice-free period of 2017. All lakes were polymictic and mixed to the bottom several times during summer (Fig. 5 f-h). Water temperatures in the surface mixed layer were lowest in Mellersta Harrsjön (9.4 ± 5.0 °C), where the mooring was placed next to the stream outlet (Fig. 1), and were higher in Inre Harrsjön (9.9 ± 5.5 °C) and Villasjön (10.2 ± 5.3 °C) (ice-free seasons of 2009–2017, mean ± SD). In early summer (June, July) deep mixing followed surface cooling and heavy rainfall. Water level maxima and surface temperature minima were observed 2-3 days after rainfall events, for example between 15 and 18 July 2017 (Fig. 5e). Strong nocturnal cooling on 16 August 2017 broke up stratification and the lakes remained well-mixed until ice-on (20 October). Increased wind speeds in September and October may have further enhanced mixing. Overall (2009–2018), stratified periods ($z_{mix}$ ≤1 m) were common (29% and 44% of the time in Inre and Mellersta Harrsjön, respectively) but were frequently interrupted by deeper mixing events. Shallow mixing ($z_{mix}$ ≤ $z_{mean}$) occurred on diel timescales. Deep mixing occurred at longer intervals (days-weeks), and more frequently toward the end of the ice-free season (Fig. 5g,h).

The gas transfer velocity generally followed the temporal pattern of the wind speed (Fig. 4b). Due to model calibration, the chamber-derived gas transfer velocities (Fig. 4b, orange rhombuses) tracked those computed with the surface renewal model (Fig. 4b, blue line). Discrepancies pointed to a mismatch between 24-hour integrated chamber fluxes and surface concentrations measured at a single point in time.  For example, measuring a low surface concentration in the de-gassed water column after a windy period during which the surface flux was high led to an overestimated $k_{ch}$ on 21 September 2017. Contrastingly, $k_{ch}$ was lower than $k_{mod}$ on 3 August 2017 due to elevated surface concentrations and a low chamber flux associated with a warm and stratified period preceding sampling. The mixed layer water temperature exceeded the air temperature by 1.6 °C on average (Fig. 5a). The bias was a function of temperatures at night dropping below surface water temperatures, which contributed to negative buoyancy fluxes at night and during cold fronts throughout the ice-free season (Fig 5b, Fig. 4i-k). We computed elevated contributions of the buoyancy flux to the TKE budget during the night and in the warmest months (Fig. 7), but the overall influence of convection on near-surface turbulence was minor. Averaged over all ice-free seasons (2009–2017) the buoyancy flux contributed only 8% to the TKE dissipation rate, but up to 90% during rare, very calm periods ($U_{10}$ ≤ 0.5 m s$^{-1}$, Fig. 4k) and up to 25% on the warmest days ($T_{surf}$ ≥ 18 °C, Fig. 4j).





**Figure 5** – Timeseries of air and mixed-layer water temperature (**a**), wind speed, gas transfer velocity from the surface renewal model ($k_{mod}$ and its buoyancy component, $k_{mod,\beta}$) and from chamber observations ($k_{ch}$) (three-lake mean values, error bars represent 95% confidence intervals) (**b**), chamber $CH_4$ flux (**c**), air-water $CH_4$ concentration difference (**d**), precipitation and changes in water level in Mellersta Harrsjön (**e**) and the water temperature in Villasjön (**f**), Inre Harrsjön (**g**) and Mellersta Harrsjön (**h**) during the ice-free season of 2017 (1 June to 20 October). The white lines in panels **e** and **f** represent the depth of the actively mixing layer. Thin and thick curves in panels **a** and **b** represent half-hourly and daily means, respectively. In panel **a** only the half-hourly timeseries of $T_{water}$ was plotted.



**Table 3** – Lake morphometry, mixing regime and $CH_4$ residence time. Mean values were calculated over the ice-free
seasons of 2009–2017.

| Lake | Area (ha) | Depth (m) | | Mixed layer depth (m) | | $N$ (cycles h$^{-1}$) | | $CH_4$ residence time (days) | |
|---|---|---|---|---|---|---|---|---|---|
| | | mean | max | mean ± SD | $n$ | mean ± SD | $n$ | mean ± SD | $n$ |
| Villasjön | 17.0 | 0.7 | 1.3 | 0.7 ± 0.3 | 66439 | 5.7 ± 8.0 | 59552 | 1.0 ± 0.4 | 72 |
| Inre Harrsjön | 2.3 | 2.0 | 5.2 | 2.5 ± 1.6 | 58362 | 5.2 ± 6.9 | 66757 | 3.4 ± 1.9 | 73 |
| Mellersta Harrsjön | 1.1 | 1.9 | 6.7 | 3.2 ± 2.9 | 62472 | 5.3 ± 9.0 | 61268 | 3.7 ± 1.7 | 72 |


### 3.3 $CH_4$ storage and residence times
Residence times of stored $CH_4$ varied between 12 hours and 7 days and were inversely correlated with
wind speed in all three lakes (OLS: $R^2 \geq 0.57$, Fig. 6). The mean residence time was shortest in the
shallowest lake, and was not significantly different between the two deeper lakes (paired t-test, $p < 0.01$,
Table 3). We did not find a statistically significant linear correlation between the residence time and day
of year or the water temperature. $CH_4$ storage was greatest in the deeper lakes and displayed patterns
similar to the surface concentrations, increasing in the warmest months with water temperature and
decreasing with wind speed.

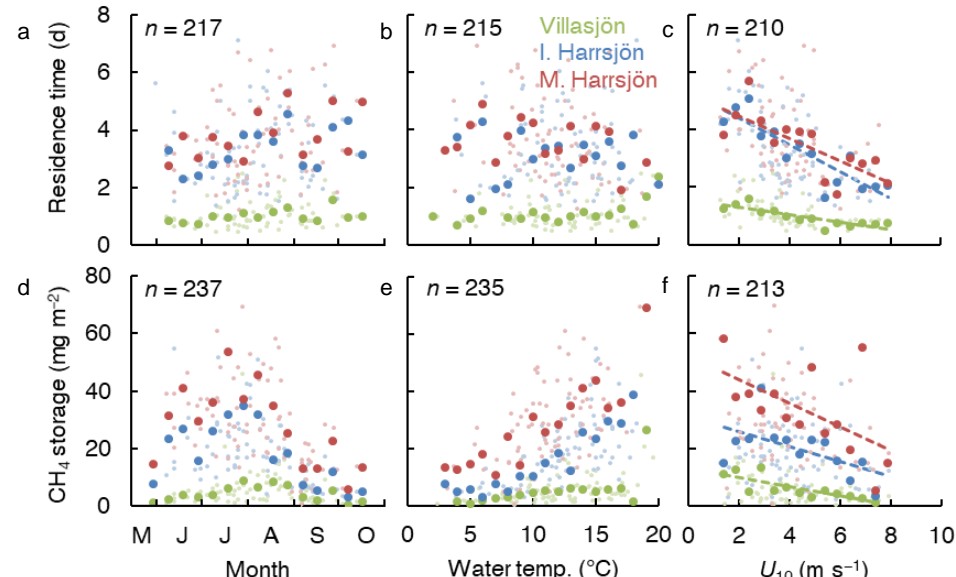

**Figure 6** – Scatterplots of the $CH_4$ residence time and storage versus time, surface water temperature
and wind speed. Symbol colours represent the different lakes. Large symbols represent binned means,
small symbols represent individual estimates. Bin sizes were 10 days, 1 °C and 0.5 m s$^{-1}$ for time, water
temperature and $U_{10}$, respectively. Linear relations of binned quantities and the wind speed were
statistically significant (residence time: $p \leq 0.002$; storage: $p \leq 0.04$). The linear regressions of the
residence time onto time of measurement and the surface water temperature were not statistically
significant ($p = 0.07–0.10$).



**3.4 Variability**

Chamber fluxes and surface water concentrations differed significantly between lakes (ANOVA, $p <$ 0.001, $n = 287$, $n = 365$). Both quantities were inversely correlated with lake surface area (Table 2). $CH_4$ concentrations in the stream feeding the Mire (22.2 ± 5.1 mg m$^{-3}$, $n = 29$, mean ± 95% CI), were significantly higher than those in the lakes (Table 2) (Lundin et al., 2013). Surface water concentrations over the deep parts of the deeper lakes (≥ 2 m water depth) were lower than those in the shallows (< 2 m) by 21 to 26% for Inre and Mellersta Harrsjön, respectively. However, the diffusive $CH_4$ flux did not differ significantly between depth zones in either Inre Harrsjön (ANOVA, $p = 0.27$, $n = 290$) or Mellersta Harrsjön (ANOVA, $p = 0.90$, $n = 293$), or between zones of high and low $CH_4$ ebullition in Villasjön (paired t-test, $p = 0.27$, $n = 89$). This is a contrast with ebullition, for which the highest fluxes were consistently observed in the shallow lake and littoral areas of the deeper lakes (Jansen et al., 2019; Wik et al., 2013).

Relations between the flux and its drivers — temperature, wind speed and the surface concentration — manifested on different timescales (Fig. 7). Over the ice-free season both the $CH_4$ fluxes and surface water concentrations tracked changes in the water temperature. The wind speed ($U_{10}$) showed less variability over seasonal (CV = 7%, $n = 17$) than over diel timescales (CV = 12%, $n = 24$) and displayed a clear diurnal maximum. The surface water/sediment temperature varied primarily on a seasonal timescale (CV = 52%/45%, $n = 17$), and less on diel timescales (CV = 3%/2%, $n = 24$). Similar to the wind speed the gas transfer velocity varied primarily on diel timescales (Fig. 7), albeit with a lower amplitude, because $k_{mod} \propto u^{3/4}$ (Eq. 4). The surface concentration correlated with wind speed and temperature (Fig. 4f,g), and showed both seasonal and diel variability. On diel timescales Δ[$CH_4$] appeared out of phase with $k_{mod}$ and peaked just before noon, when the gas transfer velocity reached its maximum value (Fig. 7b,d). However, binned means of Δ[$CH_4$] were not significantly different at the 95% confidence level (error bars) and the 1-hour chamber fluxes did not show a clear diel pattern (Fig. 7b).

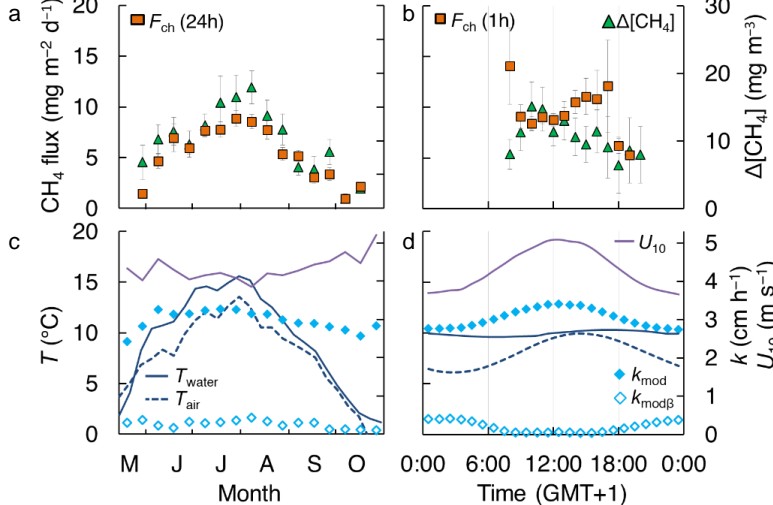

**Figure 7** – Temporal patterns of $CH_4$ chamber fluxes, concentrations (**a,b**), gas transfer velocity, air and surface water temperature and wind speed (**c,d**). Bin sizes are 10 days (**a,c**) and 1 hour (**b,d**). Error bars represent 95% confidence intervals of the binned means.





### 3.5 Timescale analysis


The spectral density plot (Fig. 8a) disentangles dominant timescales of variability of the drivers of the
flux. The power spectra of wind speed and temperature peaked at periods of 1 day and 1 year, following
well-known diel and annual cycles of insolation and seasonal variations in climate (Baldocchi et al.,
2001). For $U_{10}$, the overall spectral density maximum between 1 day and 1 week corresponds to
synoptic-scale weather variability, such as the passage of fronts (MacIntyre et al., 2009). $U_{10}$ and $T_{air}$ also
exhibit spectral density peaks at 1–3 weeks, which could be associated with persistent atmospheric
blocking typical of the Scandinavian region (Tyrlis and Hoskins, 2008). While the temperature variability
was concentrated at annual timescales, the wind speed varied primarily on timescales shorter than
about a month.

The climacogram (Fig. 8b) reveals that the variability of the chamber flux and the gas transfer velocity
was enveloped by that of the water temperature and the wind speed, as was the surface concentration
difference for timescales < 5 months. The distribution of variability over the different timescales is
similar to that shown in the spectral density plot (Fig. 8a). The standard deviation of the water
temperature did not change from its initial value ($\sigma/\sigma_{init}$ = 1) until timescales of about 1 month, following
the 1 year harmonic. In contrast, most of the variability of the wind speed was concentrated at time
scales shorter than 1 month. The variability of the chamber and modelled fluxes first tracked that of the
wind speed, but for timescales longer than about 1 month the decrease in variability resembled that of
water temperature. The variability of the modelled fluxes followed that of the surface concentration
difference rather than the gas transfer velocity. However, the coarse sampling resolution of the fluxes
and concentrations may have led to an underestimation of both the variability at <1 week timescales
(Fig. 7b) and the value of $\sigma_{init}$. Finally, the climacogram shows that $k_{mod}$ retains about 72% of its variability
at 24-hour timescales, which justifies our averaging over chamber deployment periods for comparison
with $k_{ch}$ and the computation of the model scaling parameter $\alpha'$ (Fig. 3).

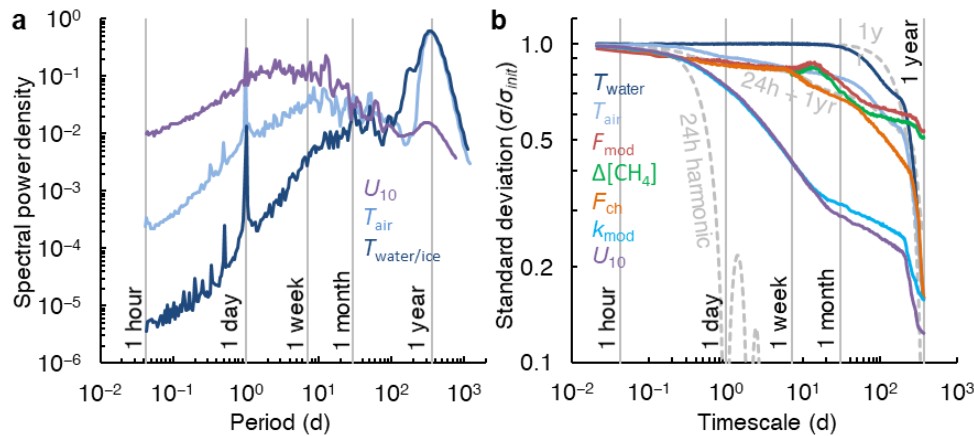


**Figure 8** – Timescale analysis of the diffusive CH$_4$ flux and its drivers. **a**: Normalized spectral density of
whole-year near-continuous timeseries of the air temperature ($T_{air}$), surface water temperature (0.1–0.5
m, $T_{water/ice}$) and the wind speed ($U_{10}$). **b**: Climacogram of the measured and modelled CH$_4$ flux ($F_{ch}$, $F_{mod}$),
the air and surface water temperature ($T_{air}$, $T_{water}$), water-air concentration difference ($\Delta[CH_4]$), modelled
gas transfer velocity ($k_{mod}$) and the wind speed ($U_{10}$) during the ice-free seasons of 2009–2017. Dashed,
light-grey curves represent (combinations of) trigonometric functions of mean 0 and amplitude 1 with a
specified period. 24h and 1yr harmonic functions were continuous over the dataset period while the 24h
+ 1yr harmonic was limited to periods when chamber flux data were available.





## 4. Discussion

### 4.1 Magnitude

Overall, diffusive emissions were lower than the average of postglacial lakes north of 50°N, but within the interquartile range (12.5, 3.0–17.9 mg m$^{-2}$ d$^{-1}$, Wik et al., 2016b). Emissions are also on the lower end of the range for northern lakes of similar size (0.01–0.2 km$^2$) (1–100 mg m$^{-2}$ d$^{-1}$, Wik et al., 2016b). As emissions of the Stordalen lakes do not appear to be limited by substrate quality or quantity (Wik et al., 2018), but strongly depend on temperature (Fig. 4b), the difference is likely because a majority of flux measurements from other postglacial lakes were conducted in the warmer, subarctic boreal zone. Boreal lake CH$_4$ emissions are generally higher for lakes of similar size: 20–40 mg m$^{-2}$ d$^{-1}$ (binned means), $n$ = 91 (Rasilo et al., 2015); ~12 mg m$^{-2}$ d$^{-1}$, $n$ = 72 (Juutinen et al., 2009).

The gas transfer velocity in the Stordalen lakes was similar to Cole and Caraco (1998) and Crusius and Wanninkhof (2003) at low wind speeds, both of which were based on tracer experiments with sampling over several days, and thus, like our approach, are integrative measures (Fig. 9). At higher winds we obtain lower $k$-values by nearly a factor of 2 (Table S1). The slope of the linear wind-$k_{ch}$ relation (OLS: 0.81 ± 0.21, slope ± 95% CI, dashed yellow line in Fig. 9) was similar to that reported by Soumis et al. (2008) (0.78 for a 0.06 km$^2$ lake), who also used a mass balance approach, and Vachon and Prairie (2013) (0.70–1.16 for lakes 0.01–0.15 km$^2$). Part of the difference with literature models was caused by the offset at 0 wind speed, which may stem from a larger contribution of the buoyancy flux (Crill et al., 1988; Read et al., 2012) or from remnant wind shear turbulence (MacIntyre et al., 2018). Another explanation may be the damping of turbulence by near-surface stratification (MacIntyre et al., 2010, 2018), however, such stratification was intermittent in our study (Fig. 5f-h). It may also result from our typically having a stable atmosphere in the day for much of the summer which reduces momentum transfer to the water surface. While our calculations take atmospheric stability into account, work on modelling momentum flux and related drag coefficients under stable atmospheres is ongoing and may lead to lower dissipation rates than we compute (Grachev et al., 2013). Due to the large spread of the chamber-derived gas transfer velocities (small rhombuses, Fig. 9) a power-law exponent to $U_{10}$ ($1.0^{1.8}_{0.0}$; exponent and 95% CI) and thus the nature of the wind-$k$ relation could not be determined with confidence.


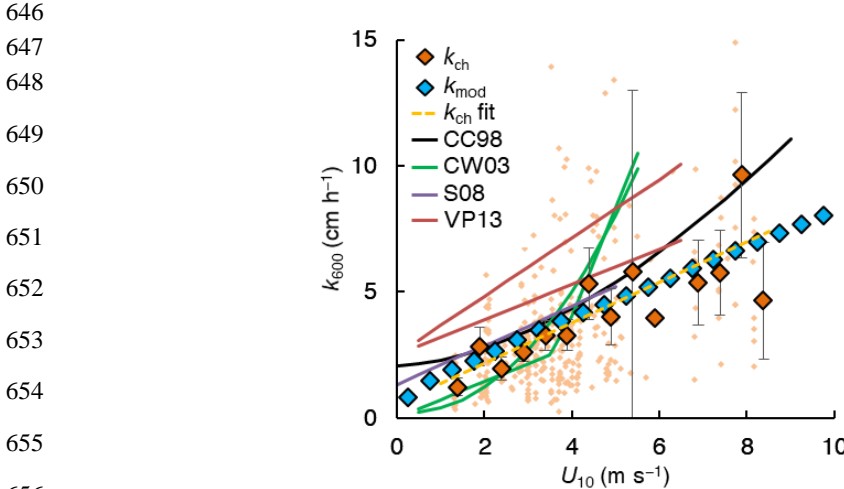

**Figure 9** – Normalized gas transfer velocities ($k_{600}$) versus the wind speed at 10 m ($U_{10}$). Binned values
(large rhombuses) and individual observations (small rhombuses) from floating chambers ($k_{ch}$) and the
surface renewal model ($k_{mod}$ with $\alpha' = 0.24$). Error bars represent 95% confidence intervals of the binned
means. Solid lines represent models from the literature: Cole and Caraco (1998), Crusius and
Wanninkhof (2003) (bilinear and power law models), Soumis et al. (2008) and Vachon and Prairie (2013)
for lake surface areas of 0.01 and 0.15 km². Supplementary Table 1 lists the model equations and
calibration ranges. A linear regression model is shown for the $k_{ch}$ data (dashed yellow line): $k_{600} =$
$0.8_{0.6}^{1.0} \times U_{10} + 0.6_{-0.2}^{+1.3}$ (sub- and superscripts denote 95% confidence intervals), with R² = 0.20 for
individual chamber values (small orange rhombuses) and R² = 0.64 for the binned means (large orange
rhombuses).



### 4.2 Drivers of the flux

The Arrhenius-type relation of $CH_4$ fluxes and concentrations (Fig. 4b,f) together with short $CH_4$ residence
times (Fig. 6) suggest that emissions from the Stordalen lakes were strongly coupled to sediment
production through efficient redistribution of dissolved $CH_4$. High $CH_4$ concentrations in the stream
suggest that terrestrial inputs of $CH_4$ may have elevated emissions in Mellersta Harrsjön (Lundin et al.,
2013; Paytan et al., 2015). Similarly, terrestrial inputs of nutrients may have indirectly enhanced
emissions in the littoral zones by supporting production of autochthonous organic substrates (Davidson
et al., 2018; Rantala et al., 2016). However, although the Mire exports substantial quantities of DOC and
presumably $CH_4$ from the water-logged fens to the lakes (Olefeldt and Roulet, 2012), after cold and rainy
periods we observed either a decrease in $\Delta[CH_4]$ (13–19 July 2017, Fig. 5) or no significant change (3–6
July and 21–27 August 2017, Fig. 5). It remains unclear whether such reduced storage resulted from
lower methanogenesis rates, convection-induced degassing or lake water displacement by surface
runoff.
Turbulent transfer was dominated by wind shear in the Stordalen lakes. We computed a minor
contribution (~8%) of the buoyancy-controlled fraction of $k$ ($k_{600,\beta}$ = 0.3 cm h$^{-1}$) (ice-free season mean,
2009–2017). Our results differ from that in Read et al. (2012) who expect a dominant role of convection
to $k$ in small lakes. The difference here results from low values of sensible and latent heat fluxes due to
colder temperatures during summer such that net long wave radiation was often less than 50 W m$^{-2}$.
Lakes in warmer regions with lower humidity and clearer skies and low wind speeds particularly at night
will have a larger contribution of buoyancy flux to the gas transfer coefficient (MacIntyre and Melack,
2009). The contribution of convection also depends on the wind-sheltering properties of the landscape
surrounding the lake (Kankaala et al., 2013; Markfort et al., 2010). Depending on the turbulence
environment the buoyancy flux is thus weighed differently in parameterizations of $\varepsilon$ (Heiskanen et al.,
2014; Tedford et al., 2014) and in wind-based models (offsets at $U_{10}$ = 0 in Fig. 9), contributing to
significant differences between model realizations of $k$ (Dugan et al., 2016; Erkkilä et al., 2018; Schilder
et al., 2016). We expect our results to be representative of small, wind-exposed lakes in cold
environments.
### 4.3 Storage and stability

The lake mixing regime can modulate flux-temperature relationships by periodically decoupling
production from emission rates (e.g. Yvon-Durocher et al., 2014). Enhanced accumulation during periods
of stratification may have contributed to concentration and storage maxima in July and August (Fig. 4e,
6d). However, as the $CH_4$ residence time was invariant over the season and with temperature (Fig. 6a,b),
the storage-temperature relation (Fig. 6e) likely reflects rate changes in sediment methanogenesis rather
than inhibited mixing. For example, the highest $CH_4$ concentrations in our dataset (59.1 ± 26.4 mg m$^{-3}$, $n$
= 37) were measured during a period with exceptionally high surface water temperatures ($T_{water}$ = 18.5 ±
3.6 °C) that lasted from 23 June to 30 July 2014. Emissions during this period comprised 29%–56%
(depending on lake) of the 2014 ice-free diffusive flux, while the peak quantity of accumulated $CH_4$ was
<5%. Two mechanisms may explain the lack of $CH_4$ accumulation. First, stratification was frequently
disrupted by vertical mixing (Fig. 5g-h) and concurrent hypolimnetic $CH_4$ concentrations were not
significantly different from (Inre Harrsjön, 2010–2017, paired t-test, $p$ = 0.12, $n$ = 32) or lower than



(Mellersta Harrsjön, 2010–2017, paired t-test, $p < 0.01$, $n = 35$) those in the surface mixed layer. Second,
stratification often was not strong enough to affect gas transfer velocities ($N>25$ during <17% of this
period). Even when assuming $\varepsilon$ was suppressed by an order of magnitude for $N>25$ and by two orders of
magnitude for $N>40$ (MacIntyre et al., 2018), $k_{mod}$ was only slightly lower (2.8 cm h$^{-1}$) than the multi-year
mean (3.0 cm h$^{-1}$). Thus, in weakly stratified, polymictic lakes, the temperature sensitivity of diffusive CH$_4$
emissions may be observed without significant modulation by stratification.
The water-air concentration difference acted as a negative feedback that maintained a quasi steady state
between CH$_4$ production and removal processes throughout the ice-free season. In other words, higher
temperatures led to elevated CH$_4$ concentrations (Fig. 4f) which in turn increased emission rates (Eq. 1,
Fig. 4b). However, in contrast to the temperature-binned fluxes, when binned by wind speed high
emission rates were associated with low concentrations (Fig. 4c,g). In this way the Δ[CH$_4$] feedback
limited the increase of the emission rate with the gas transfer velocity. In all three lakes CH$_4$ residence
times were inversely proportional to the wind speed (Fig. 6c), indicating an imbalance between
production and removal processes. We hypothesize that the imbalance exists because the variability of
wind speed peaked on shorter timescales than that of the water temperature (Fig. 8a). Changes in wind
shear periodically pushed the system out of production-emission equilibrium, allowing for transient
degassing and accumulation of dissolved CH$_4$. The temporal variability of dissolved gas concentrations is
likely higher in wind-exposed systems with limited buffer capacity (Natchimuthu et al., 2016, 2017), and
should be taken into account when applying gas transfer models to small lakes and ponds.
Rapid degassing occurred at $U_{10} > 6.5$ m s$^{-1}$ (Fig. 4c, mean wind speed during chamber deployments). Gas
fluxes at high wind speeds may have been enhanced by the kinetic action of breaking waves (Terray et
al., 1996) or through microbubble-mediated transfer. Wave breaking was observed on the Stordalen
lakes at wind speeds ≥ 7 m s$^{-1}$. Microbubbles of atmospheric gas (diameter < 1 mm) can form due to
photosynthesis, rain or wave breaking (Woolf and Thorpe, 1991) and remain entrained for several days
(Turner, 1961). Due to their relatively large surface area they quickly equilibrate with sparingly soluble
gases in the water column, providing an efficient emission pathway to the atmosphere when the bubbles
rise to the surface (Merlivat and Memery, 1983). In inland waters microbubble emissions of CH$_4$ have
only been indirectly inferred from differences in CO$_2$ and CH$_4$ gas transfer velocities (McGinnis et al.,
2015; Prairie and del Giorgio, 2013), and more work is needed to evaluate their significance in relatively
sheltered systems.
**4.4 Timescales of variability**
Overall, the short-term variability of the flux due to wind speed was similar to the long-term variability
due to temperature (ranges of the binned means, Fig. 4a-c). The diel patterns in the mixing depth (Fig. 5)
and the gas transfer velocity (Fig. 7d) and daytime variation of the surface concentration (Fig. 7b) were
indicative of daily storage-and-release cycles, resulting in a model flux difference of about 5 mg m$^{-2}$ d$^{-1}$
between morning and afternoon; about half the mean seasonal range (Fig. 7a). Diel variability of lake
methane fluxes has been observed at Villasjön (eddy covariance, Jammet et al., 2017) and elsewhere
(Bastviken et al., 2004, 2010; Crill et al., 1988; Erkkilä et al., 2018; Eugster et al., 2011; Hamilton et al.,
1994; Podgrajsek et al., 2014b). Similarly, diel patterns in the gas transfer velocity have been observed





with the eddy covariance technique (Podgrajsek et al., 2015) and in model studies (Erkkilä et al., 2018).
Apparent offsets between the diurnal peaks of the flux, surface concentrations and drivers (Fig 7b,d)
have been noted previously (Koebsch et al., 2015), but have yet to be explained. Continuous eddy
covariance measurements in lakes where the dominant emission pathway is turbulence-driven diffusion
could help characterize flux variability on short timescales (e.g. Bartosiewicz et al., 2015).

The $CH_4$ residence times (1–3 days) were not much longer than the diel timescale of vertical mixing (Fig.
5g,h). As a result, horizontal concentration gradients developed in the deeper lakes (Table 2). The 23 ±
11% concentration difference between depth zones in the deeper lakes (mean ± 95%) fits transport
model predictions of DelSontro et al. (2017) for small lakes (< 1 $km^2$) that highlight the role of outgassing
and oxidation during transport from production zones in the shallow littoral zones or the deeper
sediments (Hofmann, 2013). Concentration gradients may also have been caused by physical processes,
such as upwelling due to thermocline tilting (Heiskanen et al., 2014). Higher resolution measurements,
for example with automated equilibration systems (Erkkilä et al., 2018; Natchimuthu et al., 2016), are
needed to assess how much of the spatial and diel patterns of the $CH_4$ concentration can be explained by
physical drivers such as gas transfer and mixed layer deepening (Eugster et al., 2003; Vachon et al.,
2019), or by biological processes such as methanogenesis and microbial oxidation (Ford et al., 2002).

The distinct spectral peaks of $U_{10}$ and temperature (Fig. 8) suggest that flux dependencies on these
parameters (Fig. 4b,c) acted on different timescales. This has implications for the choice of models or
proxies of the flux in predictive analyses. For polymictic lakes and a climatology similar to that of the
Stordalen Mire (Malmer et al., 2005), temperature-based proxies (e.g. Thornton et al., 2015) would
resolve most of the variability of the ice-free diffusive $CH_4$ flux at timescales longer than a month.
Advanced gas transfer models that account for atmospheric stability and rapid variations in wind shear
are necessary to resolve the flux variability at timescales shorter than about a month. However, gas
transfer models can only deliver accurate fluxes if they are combined with measurements that capture
the full spatiotemporal variability of the surface concentration (Erkkilä et al., 2018; Hofmann, 2013;
Natchimuthu et al., 2016; Schilder et al., 2016). The short $CH_4$ residence times and diel pattern of $\Delta[CH_4]$
suggest that weekly sampling did not capture the full temporal variability of the surface concentrations.
Especially after episodes of high wind speeds and lake degassing (Fig. 4c,g), concentrations may not have
been representative of the 24-hour chamber deployment period.



**4.6 Model-chamber comparison**

Comparing gas transfer velocities from the floating chambers and the surface renewal model we find a scaling parameter value ($\alpha'$ in Eq. 4) of approximately 0.24 (Fig. 3). Its theoretical value ($\alpha$) is $\sqrt{2/15} \cong$ 0.37 (Katul et al., 2018) but empirically derived values ($\alpha'$) can vary between 0.1 and 0.7 over the range of moderate to high dissipation rates computed for the Stordalen lakes (Eq. 5: $\varepsilon = 10^{-7}-10^{-5}$ m$^2$ s$^{-3}$) (Esters et al., 2017; Wang et al., 2015 and references therein), when $\varepsilon$ is measured directly with acoustic Doppler- or particle image velocimetry and compared with independent estimates of $k$ using chambers (Gålfalk et al., 2013; Tokoro et al., 2008; Vachon et al., 2010; Wang et al., 2015), eddy covariance observations (Heiskanen et al., 2014) or the gradient flux technique (Zappa et al., 2007) and a sparingly soluble tracer, such as $CO_2$ or $SF_6$. Recent studies report a reasonable agreement between measured and modeled lake $CO_2$ fluxes if Eq. 4 and Eq. 5 are used with a multi-study mean $\alpha'$ of 0.5 (Bartosiewicz et al., 2015; Czikowsky et al., 2018; Erkkilä et al., 2018; Mammarella et al., 2015). While there is evidence for similar agreement for $CH_4$ with $\alpha' = 0.5$ (Erkkilä et al., 2018), this approach may exceed chamber-derived emissions by a factor of 2 (Bartosiewicz et al., 2015) – i.e. closer to our scaling parameter value of 0.24.

Because the physical drivers of gas exchange have been accounted for in the formulation of $k_{mod}$, chemical or biological factors that do not affect turbulence in the actively mixed layer but can limit surface exchange could be responsible for the observed variability in $\alpha'$. In most freshwater systems a significant fraction of $CH_4$ is removed through microbial oxidation at the sediment surface and in the water column (Bastviken et al., 2002). The Stordalen lakes remained oxygenated throughout the ice-free season and $CH_4$ stable isotopes indicate that between 24% (Villasjön) and 60% (Inre and Mellersta Harrsjön) of $CH_4$ in the water column was oxidized (Jansen et al., 2019). This may explain not only the low scaling parameter value compared to those found with other tracers, but also why $\alpha'$ was higher in Villasjön (0.31, $n = 67$) than in the deeper lakes (0.17–0.25, $n = 267$). However, more work is needed to establish how the oxidation effect partitioned between $CH_4$ reservoirs in the water column, where it would affect surface emissions, and the sediment. Other biogenic factors may also have impacted gas transfer, such as organic surface slicks in the 10–100 µm diffusive sublayer (Tokoro et al., 2008). Additionally, the wind speed may have been lower over the lakes than on the Mire due to the slight elevation (<1 m) of the surrounding peatland hummocks and the wind-sheltering effect of tall shrubs (*Betula nana L*, Malmer et al., 2005) on the shores of the deeper lakes (Fig. 1) (Markfort et al., 2010).

**4.7 Omitted fluxes?**

We investigated whether our chamber measurements may have missed high-quantity release from storage (Podgrajsek et al., 2014a). In stratified lakes mixed layer deepening can bring up accumulated gas, resulting in elevated surface fluxes, for example due to night time convection (Eugster et al., 2003), during autumn overturns (Encinas Fernández et al., 2014; Juutinen et al., 2009; Laurion et al., 2010; López Bellido et al., 2009) or rain events (Bartosiewicz et al., 2015; Ojala et al., 2011). Here however, >80% of the lakes' volume mixed on diel timescales and we did not observe substantial $CH_4$ accumulation over summer. Indeed, $CH_4$ concentrations within the 0.1–1 m surface layer of the deeper lakes (Table 2) were not significantly different from those at greater depth (Inre Harrsjön: 12.2 ± 2.7 mg m$^{-3}$, $n = 292$; Mellersta Harrsjön: 17.7 ± 4.9 mg m$^{-3}$, $n = 405$; means ± 95% CI). It is therefore unlikely that our chamber fluxes omitted emissions from hypolimnetic storage.





**5. Summary and conclusions**

In this study we combined a unique, multi-year dataset with a modelling approach to better understand environmental controls on turbulence-driven diffusion-limited $CH_4$ emissions from small, shallow lakes. Floating chambers estimated the seasonal mean flux at 6.9 mg m$^{-2}$ d$^{-1}$ and illustrated how the flux depended on temperature and wind speed. Wind shear controlled the gas transfer velocity while thermal convection and release from storage were minor drivers of the flux. $CH_4$ fluxes and surface concentrations fitted an Arrhenius-type temperature function ($E_a' = 0.88$–$0.97$ eV), suggesting that emissions were strongly coupled to rates of methanogenesis in the sediment. However, temperature was only an accurate proxy of the flux on averaging timescales longer than a month. On shorter timescales wind-induced variability in the gas transfer velocity, mixed layer depth, and storage decoupled production from emission rates. Transient stratification allowed for periodic $CH_4$ accumulation and resulted in an inverse relationship between wind speed and surface concentrations. In this way, the air-water concentration difference acted as a negative feedback to emissions and prevented complete degassing of the lakes, except at high wind speeds ($U_{10} \geq 6.5$ m s$^{-1}$).

Freshwater flux studies are increasingly focused on understanding mechanisms and developing proxies for use in upscaling efforts and process-based models. Our results show that the timescale of driver variability can inform the frequency of field measurements to yield representative datasets. Observations that capture the spatiotemporal variability of dissolved gas concentrations could help realize the potential of advanced gas transfer models to disentangle biogeochemical and physical flux drivers at half-hourly to interannual timescales. Linking model and field measurement approaches could uncover non-linear feedbacks, such as shallow lake degassing at high wind speeds, quantify biases associated with measurement timing and location, and constrain the applicability timescale of novel emission proxies.



**6. Data availability**
Data are available at www.bolin.su.se/data/.
**7. Author contribution**
JJ, MW and PC designed the study. Fieldwork and laboratory measurements were guided by JJ, JS and
MW. SM and AC developed the surface renewal model code. JJ performed the analyses and prepared the
manuscript with contributions from BT and SM.
**8. Competing interests**
The authors declare that they have no conflict of interest.
**9. Acknowledgements**
This work was funded by the Swedish Research Council (VR) with grants to P. Crill (#2007-4547 and
#2013-5562) and by the U.S. National Science Foundation with Arctic Natural Sciences Grants #1204267
and #1737411 to S. MacIntyre. The collection of ICOS data was funded by the Swedish Research Council
(#2015-06020). We thank the McGill University researchers (David Olefeldt, Silvie Harder and Nigel
Roulet) for the data they provided from the carbon flux tower that was supported by the Natural Science
and Engineering Research Council of Canada (# NSERC RGPIN-2017-04059). We are grateful to D.
Bastviken for validating our implementation of the chamber headspace equilibration model. We thank
the staff at the Abisko Scientific Research Station (ANS) for logistic and technical support. Noah Jansen
created the schematic of the floating chamber pair. We thank Carmody McCalley, Christoffer
Hemmingsson, Emily Pickering-Pedersen, Erik Wik, Hanna Axén, Hedvig Öste, Jacqueline Amante, Jenny
Gåling, Jóhannes West, Kaitlyn Steele, Kim Jäderstrand, Lina Hansson, Lise Johnsson, Livija Ginters,
Mathilda Nyzell, Niklas Rakos, Oscar Bergkvist, Robert Holden, Tyler Logan and Ulf Swendsén for their
help in the field.





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
