# Peer review of "Drivers of diffusive lake CH4 emissions on daily to multi-year time scales"

_Biogeosciences, 2019_

## Referee Comment (RC1) · Anonymous Referee #1 · 2 Oct 2019

L 33 : Statement "A significant portion of sediment-produced CH4 reaches the atmosphere by turbulence-driven diffusion-limited gas exchange" is misleading and term "significant" is conveniently vague. The synthesis of CH4 fluxes from inland waters given by Bastviken et al (2011) and cited by the authors provides a total diffusive flux of CH4 of 9.9 TgCH4/yr that is much smaller than the total flux of 103.3 TgCH4/yr. I suggest that authors be more specific and introduce quantitatively the importance of diffusive CH4 fluxes from inland waters.

L36: Chambers also "traditionally" capture CH4 ebullition fluxes in addition to diffusive fluxes.

L44: DelSontro et al. (2018) estimated global (and not regional as stated) CH4 emissions based on a statistical (and not "process-based" as stated) approach.

L 52: the formulation of equation (1) was given by Liss and Slater (1974) well before Wanninkhof (1992).

I have the impression that methane oxidation is the main process "that dissociate[s] production from emission rates", it's odd this is not mentioned in section L69-83.

L141-143: Can you please elaborate this section ? It's unclear how the effect of artificial enhancement of turbulence was discarded, and how the citation of the Ribas-Ribas et al. paper is relevant in this context, since this technical paper describes an apparatus to measure fluxes with chambers.

L164: It's strange that only one standard was used to calibrate the GC-FID (a multipoint calibration curve is recommended, Wilson et al. 2018), and the value of standard is so low compared to the sample values, as $pCH_4$ in the headspace was » 2 ppm, as shown in Figure 2. Authors should provide an accuracy and precision of the $CH_4$ measurements and propagate this into an error analysis of the $CH_4$ fluxes, as well as for the computed $k_{600}$ values.

L168: Could be useful to explain here how $z_{mix}$ was estimated from the temperature profiles.

L206 : This equation assumes that $C_{aq}$ remains unchanged during the 24h chamber deployment which seems unrealistic. Please clarify what does $C_{aq}$ correspond to. Was $C_{aq}$ measured each time $C_h$ was measured ?

L207: specify if T is the average during the 24h chamber deployment.

In Eq[3] explain how $dx/dt$ was computed. Linear regression over all points ? Difference between end and start ? Difference between each of the samples ?

The use of a single value for scalar $c_1$ is surprising because the accumulation of $CH_4$ in the chamber should depend on the flux intensity itself, so I would expect this value not to be constant.

In equations 2 and 3, the same symbol (T) is used for water temperature and air temperature, when separate symbols should be used for distinct variables.

L 241: It's odd that both a symbol and an abbreviation are used for turbulent kinetic energy

L307: Please explain how the residence time of a CH4 molecule in the lake was estimated.

In Figure 6 the relation between storage time and water T seems significant for I Harrjon and M Harrsjon.

L637-639: why would damping of turbulence by near-surface stratification affect particularly your lakes but not those reported by Cole & Caraco (1998) and Wanninkhof and Crusius (2003) ?

An alternative explanation could be fetch limitation (Wanninkhof 1992) in the very small sampled ponds, and this effect could be more marked at high wind speeds than at low wind speeds.

Figure 9: abbreviations given in the plot should be defined in the figure legend.

In Figure 9, the binned data value at highest wind correspond to a wind speed that is higher than highest wind speeds of individual Kch measurements. How is this possible ? The binned value should be below the highest individual wind speeds measurements.

668-670. While I agree with the idea that CH4 is formed in the sediment, as this seems the most likely process in this type of environments, I do not see why the Arrhenius relation proves this. All biological processes follow Arrhenius-type relations, so the occurrence of this relation only shows that CH4 might be biologically produced, but does not allow to pin-point it as sedimentary. Please rephrase. Since it's not explained in the text how the residence time was computed it is not clear how this proves or disproves a sedimentary CH4 production.

L671: Why do high CH4 in the stream suggest this is of "terrestrial" origin? CH4 is also produced in-stream in sediments. Do you mean that CH4 comes from soils then to streams ? or that the stream CH4 production is fueled by terrestrial organic matter ? This statement is very vague and confusing, please clarify.

L677-679: or alternatively from dilution with water with low CH4 from surface runoff and rain ?

L723: methane oxidation is also an important removal process that should contribute to imbalances between production and emission.

730-740: Wave breaking and bubbles also explain why the relation between the gas transfer velocity and wind speed is non-linear in the ocean (e.g. Wanninkhof 1992), while here you report a linear relation between gas transfer velocity and wind speed.

763 : Is thermocline tilting expected to occur in small ponds ?

797-811: Methane oxidation affects CH4 concentrations, so it's very obscure why methane oxidation should affect the alpha term. This is a scaling between gas transfer velocity that is measured and modelled, and gas transfer velocity depends on physical processes (mainly turbulence) that have nothing to do with CH4 concentration, and how it's affected by oxidation.

Refs

Liss P. S. & P. G. Slater (1974) Flux of gases across the air-sea interface, Nature, DOI: 10.1038/247181a0

Wilson et al. (2018) An intercomparison of oceanic methane and nitrous oxide measurements, Biogeosciences, 15, 5891-5907
* * *

---

## Author Comment (AC1) · 9 Oct 2019

Response to reviewer 1:

The authors wish to thank the reviewer for their thoughtful comments and insightful suggestions.

RC1: L 33: Statement "A significant portion of sediment-produced CH4 reaches the atmosphere by turbulence-driven diffusion-limited gas exchange" is misleading and term "significant" is conveniently vague. The synthesis of CH4 fluxes from inland waters given by Bastviken et al (2011) and cited by the authors provides a total diffusive flux of CH4 of 9.9 TgCH4/yr that is much smaller than the total flux of 103.3 TgCH4/yr. I suggest that authors be more specific and introduce quantitatively the importance of

diffusive CH4 fluxes from inland waters.

Author's response: we have changed the introductory paragraph to include the estimated contribution of open water diffusive CH4 emissions from three regional and global budget studies. We note that the Bastviken et al. (2011) study separates 'diffusive' and 'storage' emissions. Because the latter is defined as the 'flux when CH4 stored in the water column is emitted upon lake overturn', and occurs via the diffusion-limited pathway, we counted storage fluxes as diffusive fluxes. We thus computed the contribution of diffusion from the pathway specific budgets in Table 1 of Bastviken et al. (2011) as follows: ('diffusive' + 'storage')/('plant flux' + 'ebullition' + 'diffusive' + 'storage') = 34.8%. This is within the range of values computed from diffusive and ebullitive flux estimates in DelSontro et al. (2018) (21-24%) and Wik et al. (2016b) (46%).

RC1: L36: Chambers also "traditionally" capture CH4 ebullition fluxes in addition to diffusive fluxes.

Author's response: as described in the method section (L. 124-125) our chambers were equipped with plastic shields to prevent bubbles from entering the chamber headspace.

RC1: L44: DelSontro et al. (2018) estimated global (and not regional as stated) CH4 emissions based on a statistical (and not "process-based" as stated) approach.

Author's response: we have removed the reference to DelSontro et al. (2018).

RC1: L 52: the formulation of equation (1) was given by Liss and Slater (1974) well before Wanninkhof (1992).

Author's response: the reference has been changed.

RC1: I have the impression that methane oxidation is the main process "that dissociate[s] production from emission rates", it's odd this is not mentioned in section L69-83.

Author's response: we have included oxidation as one of the dissociating factors.

RC1: L141-143: Can you please elaborate this section? It's unclear how the effect of

artificial enhancement of turbulence was discarded, and how the citation of the Ribas-Ribas et al. paper is relevant in this context, since this technical paper describes an apparatus to measure fluxes with chambers.

Author's response: we included a more detailed description of the analysis of the cited paper: "Ribas-Ribas et al. (2018) compared acoustic Doppler velocimeter measurements inside and outside the perimeter of a chamber of similar design, size and flotation depth as those used in this study, and, based on a comparison of measured TKE dissipation rates and computed gas transfer velocities, concluded that the chambers did not cause artificial turbulence."

RC1: L164: It's strange that only one standard was used to calibrate the GC-FID (a multipoint calibration curve is recommended, Wilson et al. 2018), and the value of standard is so low compared to the sample values, as pCH4 in the headspace was Âż 2 ppm, as shown in Figure 2. Authors should provide an accuracy and precision of the CH4 measurements and propagate this into an error analysis of the CH4 fluxes, as well as for the computed k600 values.

Author's response: detector FID's with N2 carriers are known to be linear over several orders of magnitude (e.g. Colson, 1986). The linearity of the detector is better than the uncertainty in the gas mixtures. The instrument precision is discussed in Section 2.4 of the paper. 10 standard measurements before and after each run were used to assess instrument precision and drift. The precision – defined as the relative standard deviation of the 10 standard measurements - was generally <0.25%. This converts to negligible deviations in the surface concentration and derived fluxes, and would not affect any of the binned or multi-year mean values or functional relationships discussed in the paper. For example, relative standard deviations of the air-water concentration difference binned by time, temperature and wind speed (Fig. 4e-g) are generally >30%. Thus, uncertainty in this study is dominated by the spatiotemporal variability of the fluxes and surface concentrations rather than uncertainty in the concentration measurements.

RC1: L168: Could be useful to explain here how zmix was estimated from the temperature profiles.

Author's response: the mixing depth was estimated from a density gradient threshold, as described at L. 431. We have now written a few sentences about water density calculations and mixing depth in section 2.5. Here is the revised text: "Water density was computed from temperature and salinity (Chen and Millero, 1977), using lake-averaged specific conductivity and a salinity factor [mS cm−1 / g kg−1] of 0.57. The salinity factor was based on a linear regression of simultaneous measurements of conductivity and dissolved solids (R2 = 0.99, n = 7) in five lakes in the Torneträsk catchment (Miljödata-MVM, 2017). We defined the diel mixing depth (zmix) at a density gradient threshold (d/dz) of 0.03 kg m−3 m−1 (Rueda et al., 2007)."

RC1: L206: This equation assumes that Caq remains unchanged during the 24h chamber deployment which seems unrealistic. Please clarify what does Caq correspond to. Was Caq measured each time Ch was measured?

Author's response: Caq is defined at L. 51 and corresponds to the measured CH4 concentration in the surface water. When we compute gas transfer velocities from chamber fluxes and the air-water concentration difference (Caq−Cair,eq), we only use water samples that were collected simultaneously with, and in close proximity to the floating chamber observations. We took one water sample at each chamber location. Thus when we compute kch from Eq. 2, we assume that the flux at the time of the concentration measurement was equal to the 24h flux. It is likely that the flux varied over those 24 hours (Fig. 7b,d). However, a quantitative bias assessment would require continuous, 24h observations of the diffusive fluxes and of the surface concentration, which were unavailable for this study.

RC1: L207: specify if T is the average during the 24h chamber deployment. In Eq [3] explain how dx/dt was computed. Linear regression over all points? Difference between end and start? Difference between each of the samples?

[Figure]

Author's response: we have now specified how T and dx/dt were computed. That is, the average over the flux integration time (this is the 24h chamber deployment time for most of our analyses) and OLS linear regression of concentrations onto time, respectively. Here is the revised text: "$\partial xh/\partial t$ is the headspace mole fraction change [$10-6$ ppm $d-1$] computed with an ordinary least squares (OLS) linear regression (Fig. 2), M is the molar mass of $CH_4$ (0.016 mg mol$-1$), P is the air pressure [Pa], Tair is the air temperature [K]. Scalar $c\_1$ corrects for accumulation of $CH_4$ gas in the chamber headspace and increases over the deployment time. Comparing both chamber flux calculation methods we find $c\_1 = 1.21$ for 24 hour deployments (OLS, R2 = 0.85, n = 357). Chambers were sampled up to 4 times during deployment (at 10 minutes, 1–5 hours and 24 hours) which allowed us to compute fluxes at time intervals of 1 hour and 24 hours. P and Tair were averaged over the relevant time interval."

RC1: The use of a single value for scalar c1 is surprising because the accumulation of $CH_4$ in the chamber should depend on the flux intensity itself, so I would expect this value not to be constant.

Author's response: c1 is based on a linear regression of fluxes computed with Eq. 3 (simple linear regression in the time vs chamber headspace concentration plot) and Eq. 2, which corrects for the headspace effect. The good linear fit (R2 = 0.85, L. 216) indicates to us that the headspace effect did not change significantly within the range of fluxes observed. If the headspace accumulation effect would increase significantly with the flux, we would expect a highly non-linear correlation.

RC1: In equations 2 and 3, the same symbol (T) is used for water temperature and air temperature, when separate symbols should be used for distinct variables.

Author's response: we have specified the symbols in the equations.

RC1: L 241: It's odd that both a symbol and an abbreviation are used for turbulent kinetic energy

Author's response: they refer to different quantities. As stated at L. 241, we use TKE as an abbreviation for 'turbulent kinetic energy' and epsilon as the symbol for the dissipation rate of turbulent kinetic energy. Both the abbreviation and the symbol are commonly used in the literature.

RC1: L307: Please explain how the residence time of a CH4 molecule in the lake was estimated.

Author's response: this computation is described in section 2.7 at L198-199. The sentence reads: "We computed the average residence time of a CH4 molecule by dividing the amount stored by the lake mean surface flux."

RC1: In Figure 6 the relation between storage time and water T seems significant for I Harrjon and M Harrsjon.

Author's response: we infer that the reviewer refers to Fig. 6e, which shows an increase of storage with water temperature. We fitted Arrhenius functions, added lines of best fit to the plot and included the following sentence in the figure caption: "Arrhenius-type functions (Eq. 7) adequately described the relation between storage and temperature in each lake ($R^2 \geq 0.70$, $p < 0.001$)."

RC1: L637-639: why would damping of turbulence by near-surface stratification affect particularly your lakes but not those reported by Cole & Caraco (1998) and Wanninkhof and Crusius (2003)?

Author's response: we moved the discussion of turbulence damping to a separate paragraph, which now reads: "Damping of turbulence results from near-surface stratification and can reduce the gas transfer velocity (MacIntyre et al., 2010, 2018), however, such stratification was intermittent in our study (Fig. 5f-h). It may also result from our typically having a stable atmosphere in the day for much of the summer which reduces momentum transfer to the water surface."

RC1: An alternative explanation could be fetch limitation (Wanninkhof 1992) in the very

small sampled ponds, and this effect could be more marked at high wind speeds than at low wind speeds.

Author's response: we thank the reviewer for this suggestion.

RC1: Figure 9: abbreviations given in the plot should be defined in the figure legend.

Author's response: abbreviations of literature references are now included in the figure caption.

RC1: In Figure 9, the binned data value at highest wind correspond to a wind speed that is higher than highest wind speeds of individual Kch measurements. How is this possible? The binned value should be below the highest individual wind speeds measurements.

Author's response: we chose to plot the symbols for binned quantities at the center value of each bin. The center value of the bin may be higher than or lower than the mean value of the datapoints contained in that bin.

RC1: L668-670: While I agree with the idea that CH4 is formed in the sediment, as this seems the most likely process in this type of environments, I do not see why the Arrhenius relation proves this. All biological processes follow Arrhenius-type relations, so the occurrence of this relation only shows that CH4 might be biologically produced, but does not allow to pin-point it as sedimentary. Please rephrase. Since it's not explained in the text how the residence time was computed it is not clear how this proves or disproves a sedimentary CH4 production.

Author's response: we explain how the residence time was computed at L. 198-199. Sediment production of CH4 is well-known in aquatic systems and we do not mean to prove or disprove it with our observations. We restructured the sentence to put more emphasis on redistribution rather than sediment production: "The Arrhenius-type relation of CH4 fluxes and concentrations (Fig. 4b,f) together with short CH4 residence times (Fig. 6) suggest that efficient redistribution of dissolved CH4 strongly coupled

emissions from the Stordalen lakes to sediment production."

RC1: L671: Why do high CH4 in the stream suggest this is of "terrestrial" origin? CH4 is also produced in-stream in sediments. Do you mean that CH4 comes from soils then to streams? Or that the stream CH4 production is fueled by terrestrial organic matter? This statement is very vague and confusing, please clarify.

Author's response: we have clarified this sentence as follows: "High CH4 concentrations in the stream suggest that external inputs of CH4 — produced in the fens and transported into the stream with surface runoff, or produced in stream sediments — may have elevated emissions in Mellersta Harrsjön (Lundin et al., 2013; Paytan et al., 2015)."

RC1: L677-679: or alternatively from dilution with water with low CH4 from surface runoff and rain?

Author's response: we thank the reviewer for this suggestion.

RC1: L723: methane oxidation is also an important removal process that should contribute to imbalances between production and emission.

Author's response: One could argue that because methane oxidation rates tend to change with concentration and temperature, they would influence the flux on timescales similar to those of production (that is, timescales of a week or more). Changes in storage occur within the short residence times of CH4 gas (1-5 days). This suggest that dissociation occurs on shorter timescales – i.e. those governed by wind speed. However, in this paper we do not present a quantitative assessment of methane oxidation. Following the reviewer's earlier comment we have mentioned oxidation as a process that dissociates production from emission in the introduction.

RC1: L730-740: Wave breaking and bubbles also explain why the relation between the gas transfer velocity and wind speed is non-linear in the ocean (e.g. Wanninkhof 1992), while here you report a linear relation between gas transfer velocity and wind

speed.

Author's response: the processes obtained in large water bodies are not necessarily operative in small lakes. Moreover, it is not clear from our data whether the wind-k relationship is linear or non-linear. At L. 643-645 we state "Due to the large spread of the chamber-derived gas transfer velocities (small rhombuses, Fig. 9) a power-law exponent to U10 (1.0 (0.0-1.8); exponent and 95% CI) and thus the nature of the wind-k relation could not be determined with confidence."

RC1: L763: Is thermocline tilting expected to occur in small ponds?

Yes, wind forcing can cause the thermocline to tilt in small water bodies. The extent of tilt is computed from the Wedderburn number (Imberger and Patterson, 1989). This dimensionless index takes into account stratification, wind speed, and basin dimensions. The lakes in this study are larger than ponds, albeit small, and for the relatively high winds found at these arctic sites, thermocline tilting is expected.

RC1: L797-811: Methane oxidation affects CH4 concentrations, so it's very obscure why methane oxidation should affect the alpha term. This is a scaling between gas transfer velocity that is measured and modelled, and gas transfer velocity depends on physical processes (mainly turbulence)

Author's response: in this study we infer kch from measurements of the chamber concentration increase and surface concentrations. The formulations of Eq. 1 and Eq. 2 implicitly assume that all CH4 measured in the water column is emitted to the atmosphere. However, if a fraction of CH4 is removed by oxidation, this would lead to an overestimation of $\Delta$[CH4] and an underestimation of kch. This in turn affects the alpha term. So oxidation does not impact gas transfer velocities directly, but may bias the gas transfer velocity high if one uses the two-layer model (Eq. 1), as is common. We have added a few sentences to the paragraph to clarify this point.

References cited in author's responses: Bastviken, D., Tranvik, L. J., Downing, J. A.,

[Figure]

Crill, P. M. and Enrich-Prast, A.: Freshwater Methane Emissions Offset the Continental Carbon Sink, Science (80-. )., 331(6013), 50–50, doi:10.1126/science.1196808, 2011.

Chen, C.-T. and Millero, F. J.: The use and misuse of pure water PVT properties for lake waters, Nature, 266(5604), 707–708, doi:10.1038/266707a0, 1977.

Colson, E. R.: Flame ionization detectors and high-end linearity, Anal. Chem., 58(2), 337–344, doi:10.1021/ac00293a017, 1986.

Imberger, J. and Patterson, J. C.: Physical Limnology, in Advances in Applied Mechanics, vol. 27, pp. 303–475., 1989.

Lundin, E. J., Giesler, R., Persson, A., Thompson, M. S. and Karlsson, J.: Integrating carbon emissions from lakes and streams in a subarctic catchment, J. Geophys. Res. Biogeosciences, 118(3), 1200–1207, doi:10.1002/jgrg.20092, 2013.

MacIntyre, S., Jonsson, A., Jansson, M., Aberg, J., Turney, D. E. and Miller, S. D.: Buoyancy flux, turbulence, and the gas transfer coefficient in a stratified lake, Geophys. Res. Lett., 37(24), n/a-n/a, doi:10.1029/2010GL044164, 2010.

MacIntyre, S., Crowe, A. T., Cortés, A. and Arneborg, L.: Turbulence in a small arctic pond, Limnol. Oceanogr., 63(6), 2337–2358, doi:10.1002/lno.10941, 2018.

Miljödata-MVM. Swedish University of Agricultural Sciences (SLU). National data host for lakes and watercourses, and national data host for agricultural land, http://miljodata.slu.se/mvm/ [07-10-2017]. Paytan, A., Lecher, A. L., Dimova, N., Sparrow, K. J., Kodovska, F. G.-T., Murray, J., Tulaczyk, S. and Kessler, J. D.: Methane transport from the active layer to lakes in the Arctic using Toolik Lake, Alaska, as a case study, Proc. Natl. Acad. Sci., 112(12), 201417392, doi:10.1073/pnas.1417392112, 2015.

Ribas-Ribas, M., Kilcher, L. F. and Wurl, O.: Sniffle: a step forward to measure in situ CO2 fluxes with the floating chamber technique, Elem Sci Anth, 6(1), 14, doi:10.1525/elementa.275, 2018.

Rueda, F., Moreno-Ostos, E. and Cruz-Pizarro, L.: Spatial and temporal scales of transport during the cooling phase of the ice-free period in a small high-mountain lake, Aquat. Sci., 69(1), 115–128, doi:10.1007/s00027-006-0823-8, 2007.

---

## Referee Comment (RC2) · Anonymous Referee #2 · 22 Oct 2019

This manuscript documents almost a decade of weekly-monthly resolution methane concentration and flux data from 3 sub-Arctic lakes. They found Arrhenius-type temperature relationships with flux and concentration, which has been found before and suggests a strong coupling to methane production rates. They also found that wind shear drove the gas transfer velocity, but on timescales of less than a month while temperature was a driver on timescales longer than a month. They also found that stratification only played a small role in storage/accumulation and emissions in general from their systems. The methods are sound and the results are well-detailed, perhaps a bit on the long side. The dataset is quite unique as it is so long. The authors need to use the length of their dataset to substantiate their results more. They find a temperature relationship that has been shown before in quite a few other datasets, but perhaps

ones not as long as theirs. Also, they find that convection does not play as large of a role in surface turbulence as has been found in other lakes. How do those datasets compare to theirs? I also strongly suggest the authors structure the discussion to highlight the main takeaway messages from this work.

General comments: 1. The title seems broad as if you are referring to all lakes, but you actually point out in the manuscript many differences between your findings and those of other lakes, for example, in terms of convection contribution to k. I suggest you narrow down your title slightly. You could even highlight more in the title the amount of data that you have. This multi-year dataset is quite unique. 2. I think the discussion could do with some restructuring and more concisely define the main points of your findings. The subheadings closely follow the results structure, but this doesn't help the reader easily identify your main points. I like the way you summarized your findings in the first paragraph of the last section (summary and conclusions). I would suggest laying out the discussion with subheadings similar to the structure in that paragraph, at least to start and then edit from there. You also may not need all the information in the discussion if you find it does not highlight one of your main points.

Specific comments: Line 50- should read ', of which the upper boundary..'

Line 72 – did you not include Aben et al. 2017 because it is about ebullition? You don't specifically mention diffusive only in this sentence.

Line 101 – 'stochastic tools' sounds too vague here

Line 129 – I would say 'During the 24 hr period. . .' to avoid confusion. But why 2-4 samplings? What resolution and why?

Line 135-136 – you need to define Fch,unsh and Fch,sh here in this sentence (i.e., place the variables after 'shielded' and 'unshielded')

Section 2.3 – Do you flush the chambers between samplings or leave them the entire 24 hrs? Do you flush or mix the 4m long tube before sampling?

[Figure]

Line 196 – do you mean 'offshore' instead of 'nearshore' here since you are differentiating between the littoral zone and another zone?

Line 198 – make sure the year is correct on the reference

Line 205 – define and give units for 'kch'

Line 211-212 – why were there some water measurements not taken and which ones and how many?

Line 239 – should be 'kmod' specifically in this sentence, no?

Line 245 – why do you need to do this qualitative comparison? Why is it important?

Line 338 – definte '$\sigma$init'

Line 420 – include in the caption the panel letters for the histograms in parentheses too

Figure 4 caption – you need to describe the squares, triangles, and diamonds in the caption itself – all the variables that you are presenting here.

Figure 5 caption – what are the curves you speak of in line 500? Are you sure that e and f are the right panels when you discus the white lines on line 499? What is the resolution in panels c and d?

Table 3 title – need to describe N here

Figure 6 caption – add '(a-c)' after 'residence time' and '(d-f)' after 'storage'. You mention the regressions for residence time but not for storage. Also, it looks as if there could be a trend between temperature and storage (panel e) for at least 2 of the lakes. Was there not?

Line 560-561 – the sentence starting with 'On diel timescales..' needs rewording. I don't understand it.

Figure 7 – put a complete legend in panels a and c and state that they apply to panels

b and d.

Line 612 – what is 'Twater/ice'?

Section 4.1 – The subheading 'Magnitude' doesn't explain much. Magnitude of what?

Line 632 – you obtained lower k-values by nearly a factor of 2 compared to what?

Line 636 – who had the offset at 0 wind speed? You or the literature? Be specific as this sentence is a bit confusing.

Line 637 – 'Another explanation' for what?

Line 639-640 – how was the atmosphere stable?

Line 644-645 – I am confused because you have an equation in Figure 9 caption that has an exponent for u10 with 95% Cis.

Section 4.2 – delete 'the' in the subheading

Section 4.2 – this is a very important part of the discussion but I feel it needs a little more work to really bring out your main points. It reads a bit like a bunch of ideas thrown into a paragraph but without linking them all together nor highlighting why these ideas matter. For example, the first sentence states that the temperature relationship with flux and concentration suggests a strong coupling to sediment [methane] production (need that word 'methane' in there). I agree with this statement and it's an important one because you did find some nice relationships there. But the next sentence talks about stream inputs (from your own data, correct?) and then the following sentence is back to how sediment methane production could be enhanced. They seem out of order. Then the last thought about the decrease in CH4 after cold rain events is actually still in line with the temperature relationship you saw but you start this sentence off attempting to state that that shouldn't be the case if there was runoff from fens. This fens part goes more along with the streams sentence from above. I feel the same for the second paragraph of the section. I think you clarify your point about the difference

between your results and those of Read et al. I am actually not sure who had lake in the warmer, lower humidity regions – you or them? Also need to put the 50 w/m2 value in context. At the end, I wouldn't use the word 'expect' because I think you showed this. And I believe in this whole section you should already elude to the fact that these drivers work on different timescales.

Line 716-728 – The first sentence of this paragraph reads more like a summary sentence. It's confusing to hear about the feedback before you describe how you got to that point. I would try restructuring this paragraph a bit. I would start with the second sentence and state it like so: 'Higher temperatures led to elevated CH4 concentrations, which in turn increased emission rates, but high wind speed was correlated with high emission rates and low concentrations. In this way,. . .'

Line 744 – add the range of binned means in those parentheses of $\sim 0 - 10$

Line 784-791 – This is actually one very long sentence. Consider splitting it.

Line 798-799 – missing a word or something here ' . . ..but can limit surface exchange could be responsible. . .'

Line 834-837 – So you don't completely degas the lake, despite shallowness and frequent mixing, but you also don't have storage/accumulation of methane. I am finding a hard time reconciling those two results. I feel this needs more explanation here but also in the discussion where you mention it.

---

## Author Response (AR1)

January 2020

Biogeosciences Editorial Office

Manuscript ID: bg-2019-322

Dear Prof. Abril,

Thank you for taking into consideration our manuscript for publication in *Biogeosciences*. We are grateful for the granted extension of the submission deadline of our revised manuscript. This has allowed us to improve the depth of the discussion and the scientific significance of the work. We attach our responses to the reviewer's comments, as well as a revised version of the manuscript and supplementary material in which the edits have been highlighted.

We have made several important changes to the manuscript. Foremost, we have changed the manuscript title and improved the structure of the discussion, as per the suggestions of yourself and Referee #2. We have also increased the relevance of the manuscript to other researchers working on trace gas emissions from lakes by expanding the discussion with a detailed analysis of between-lake differences in the drivers of emissions, which include the effects of atmospheric stability and sheltering on the gas transfer velocity. Contextualizing in this way, we have illustrated which flux-driving mechanisms may be important in different lake types, such as those that are shallow and exposed to wind, deeper and more sheltered, or lakes that are fed by a stream. Minor changes include improved model estimates of lake $CH_4$ emissions by using lake-specific scaling parameters – the analyses for which are provided in the updated supplement.

We hope that these revisions have made our manuscript suitable for publication in *Biogeosciences*. We will be pleased to answer any additional questions or make any further changes that you or the reviewers recommend. Thank you for your kind support.

On behalf of the co-authors,

Joachim Jansen

**Response to reviewer 1:**

L 33: Statement "A significant portion of sediment-produced CH4 reaches the atmosphere by turbulence-driven diffusion-limited gas exchange" is misleading and term "significant" is conveniently vague. The synthesis of CH4 fluxes from inland waters given by Bastviken et al (2011) and cited by the authors provides a total diffusive flux of CH4 of 9.9 TgCH4/yr that is much smaller than the total flux of 103.3 TgCH4/yr. I suggest that authors be more specific and introduce quantitatively the importance of diffusive CH4 fluxes from inland waters.

Author's response: we have changed the introductory paragraph to include the estimated contribution of open water diffusive CH4 emissions from three regional and global budget studies. We note that the Bastviken et al. (2011) study separates 'diffusive' and 'storage' emissions. Because the latter is defined as the 'flux when CH4 stored in the water column is emitted upon lake overturn', and occurs via the diffusion-limited pathway, we counted storage fluxes as diffusive fluxes. We thus computed the contribution of diffusion from the pathway specific budgets in Table 1 of Bastviken et al. (2011) as follows: ('diffusive' + 'storage')/('plant flux' + 'ebullition' + 'diffusive' + 'storage') = 34.8%. This is within the range of values computed from diffusive and ebullitive flux estimates in DelSontro et al. (2018) (21-24%) and Wik et al. (2016b) (46%).

L36: Chambers also "traditionally" capture CH4 ebullition fluxes in addition to diffusive fluxes.

Author's response: as described in the method section (L. 124-125) our chambers were equipped with plastic shields to prevent bubbles from entering the chamber headspace.

L44: DelSontro et al. (2018) estimated global (and not regional as stated) CH4 emissions based on a statistical (and not "process-based" as stated) approach.

Author's response: we have removed the reference to DelSontro et al. (2018).

L 52: the formulation of equation (1) was given by Liss and Slater (1974) well before Wanninkhof (1992).

Author's response: the reference has been changed.

I have the impression that methane oxidation is the main process "that dissociate[s] production from emission rates", it's odd this is not mentioned in section L69-83.

Author's response: we have included oxidation as one of the dissociating factors.

L141-143: Can you please elaborate this section? It's unclear how the effect of artificial enhancement of turbulence was discarded, and how the citation of the Ribas-Ribas et al. paper is relevant in this context, since this technical paper describes an apparatus to measure fluxes with chambers.

Author's response: we included a more detailed description of the analysis of the cited paper: "Ribas-Ribas et al. (2018) compared acoustic Doppler velocimeter measurements inside and outside the perimeter of a chamber of similar design, size and flotation depth as those used in this study, and, based on a comparison of measured TKE dissipation rates and computed gas transfer velocities, concluded that the chambers did not cause artificial turbulence."

L164: It's strange that only one standard was used to calibrate the GC-FID (a multipoint calibration curve is recommended, Wilson et al. 2018), and the value of standard is so low compared to the sample values, as pCH4 in the headspace was » 2 ppm, as shown in Figure 2.
Authors should provide an accuracy and precision of the CH4 measurements and propagate this into an error analysis of the CH4 fluxes, as well as for the computed k600 values.

Author's response: detector FID's with N2 carriers are known to be linear over several orders of magnitude (e.g. Colson, 1986). The linearity of the detector is better than the uncertainty in the gas mixtures.

The instrument precision is discussed in Section 2.4 of the paper. 10 standard measurements before and after each run were used to assess instrument precision and drift. The precision – defined as the relative standard deviation of the 10 standard measurements - was generally <0.25%. This converts to negligible deviations in the surface concentration and derived fluxes, and would not affect any of the binned or multi-year mean values or functional relationships discussed in the paper. For example, relative standard deviations of the air-water concentration difference binned by time, temperature and wind speed (Fig. 4e-g) are generally >30%. Thus, uncertainty in this study is dominated by the spatiotemporal variability of the fluxes and surface concentrations rather than uncertainty in the concentration measurements.

L168: Could be useful to explain here how zmix was estimated from the temperature profiles.

Author's response: the mixing depth was estimated from a density gradient threshold, as described at L. 431. We have now written a few sentences about water density calculations and mixing depth in section 2.5. Here is the revised text:

"Water density was computed from temperature and salinity (Chen and Millero, 1977), using lake-averaged specific conductivity and a salinity factor [mS cm$^{-1}$ / g kg$^{-1}$] of 0.57. The salinity factor was based on a linear regression of simultaneous measurements of conductivity and dissolved solids ($R^2$ = 0.99, n = 7) in five lakes in the Torneträsk catchment (Miljödata-MVM, 2017). We defined the diel mixing depth ($z_{mix}$) at a density gradient threshold (d$\rho$/dz) of 0.03 kg m$^{-3}$ m$^{-1}$ (Rueda et al., 2007)."

L206: This equation assumes that Caq remains unchanged during the 24h chamber deployment which seems unrealistic. Please clarify what does Caq correspond to. Was Caq measured each time Ch was measured?

Author's response: $C_{aq}$ is defined at L. 51 and corresponds to the measured $CH_4$ concentration in the surface water. When we compute gas transfer velocities from chamber fluxes and the air-water concentration difference ($C_{aq}$–$C_{air,eq}$), we only use water samples that were collected simultaneously with, and in close proximity to the floating chamber observations. We took one water sample at each chamber location. Thus when we compute $k_{ch}$ from Eq. 2, we assume that the flux at the time of the concentration measurement was equal to the 24h flux. It is likely that the flux varied over those 24 hours (Fig. 7b,d). However, a quantitative bias assessment would require continuous, 24h observations of the diffusive fluxes and of the surface concentration, which were unavailable for this study.

L207: specify if T is the average during the 24h chamber deployment. In Eq [3] explain how dx/dt was computed. Linear regression over all points? Difference between end and start? Difference between each of the samples?

Author's response: we have now specified how T and dx/dt were computed. That is, the average over the flux integration time (this is the 24h chamber deployment time for most of our analyses) and OLS linear regression of concentrations onto time, respectively. Here is the revised text:

"$\partial x_h/\partial t$ is the headspace mole fraction change [mol mol$^{-1}$ d$^{-1}$] computed with an ordinary least squares (OLS) linear regression (Fig. 2), $M$ is the molar mass of CH$_4$ (0.016 mg mol$^{-1}$), $P$ is the air pressure [Pa], $T_{air}$ is the air temperature [K]. Scalar $c_1$ corrects for accumulation of CH$_4$ gas in the chamber headspace and increases over the deployment time. Comparing both chamber flux calculation methods we find $c_1$ = 1.21 for 24 hour deployments (OLS, R$^2$ = 0.85, $n$ = 357). Chambers were sampled up to 4 times during deployment (at 10 minutes, 1–5 hours and 24 hours) which allowed us to compute fluxes at time intervals of 1 hour and 24 hours. $P$ and $T_{air}$ were averaged over the relevant time interval."

The use of a single value for scalar c1 is surprising because the accumulation of CH4 in the chamber should depend on the flux intensity itself, so I would expect this value not to be constant.

Author's response: c1 is based on a linear regression of fluxes computed with Eq. 3 (simple linear regression in the time vs chamber headspace concentration plot) and Eq. 2, which corrects for the headspace effect. The good linear fit (R$^2$ = 0.85, L. 216) indicates to us that the headspace effect did not change significantly within the range of fluxes observed. If the headspace accumulation effect would increase significantly with the flux, we would expect a highly non-linear correlation.

In equations 2 and 3, the same symbol (T) is used for water temperature and air temperature, when separate symbols should be used for distinct variables.

Author's response: we have specified the symbols in the equations.

L 241: It's odd that both a symbol and an abbreviation are used for turbulent kinetic energy

Author's response: they refer to different quantities. As stated at L. 241, we use TKE as an abbreviation for 'turbulent kinetic energy' and epsilon as the symbol for the dissipation rate of turbulent kinetic energy. Both the abbreviation and the symbol are commonly used in the literature.

L307: Please explain how the residence time of a CH4 molecule in the lake was estimated.

Author's response: this computation is described in section 2.7 at L198-199. The sentence reads: "We computed the average residence time of a CH$_4$ molecule by dividing the amount stored by the lake mean surface flux."

In Figure 6 the relation between storage time and water T seems significant for I Harrjon and M Harrsjon.

Author's response: we infer that the reviewer refers to Fig. 6e, which shows an increase of storage with water temperature. We fitted Arrhenius functions, added lines of best fit to the plot and included the following sentence in the figure caption: "Arrhenius-type functions (Eq. 7) adequately described the relation between storage and temperature in each lake (R$^2$ ≥ 0.70, $p$ < 0.001)."

L637-639: why would damping of turbulence by near-surface stratification affect particularly your lakes but not those reported by Cole & Caraco (1998) and Wanninkhof and Crusius (2003)?

Author's response: we moved the discussion of turbulence damping to a separate paragraph, which now reads: "Damping of turbulence results from near-surface stratification and can reduce the gas transfer velocity (MacIntyre et al., 2010, 2018), however, such strong stratification (N > 25 cph) was intermittent in our study (Fig. 5f-h)."

An alternative explanation could be fetch limitation (Wanninkhof 1992) in the very small sampled ponds, and this effect could be more marked at high wind speeds than at low wind speeds.

Author's response: we thank the reviewer for this suggestion.

Figure 9: abbreviations given in the plot should be defined in the figure legend.

Author's response: abbreviations of literature references are now included in the figure caption.

In Figure 9, the binned data value at highest wind correspond to a wind speed that is higher than highest wind speeds of individual Kch measurements. How is this possible? The binned value should be below the highest individual wind speeds measurements.

Author's response: we chose to plot the symbols for binned quantities at the center value of each bin. The center value of the bin may be higher than or lower than the mean value of the datapoints contained in that bin.

L668-670: While I agree with the idea that CH4 is formed in the sediment, as this seems the most likely process in this type of environments, I do not see why the Arrhenius relation proves this. All biological processes follow Arrhenius-type relations, so the occurrence of this relation only shows that CH4 might be biologically produced, but does not allow to pin-point it as sedimentary. Please rephrase. Since it's not explained in the text how the residence time was computed it is not clear how this proves or disproves a sedimentary CH4 production.

Author's response: we explain how the residence time was computed at L. 198-199. Sediment production of CH4 is well-known in aquatic systems and we do not mean to prove or disprove it with our observations. We restructured the sentence to put more emphasis on redistribution rather than sediment production: "The Arrhenius-type relation of $CH_4$ fluxes and concentrations (Fig. 4b,f) together with short $CH_4$ residence times (Fig. 6) suggest that efficient redistribution of dissolved $CH_4$ strongly coupled emissions from the Stordalen lakes to sediment production."

L671: Why do high CH4 in the stream suggest this is of "terrestrial" origin? CH4 is also produced in-stream in sediments. Do you mean that CH4 comes from soils then to streams? Or that the stream CH4 production is fueled by terrestrial organic matter? This statement is very vague and confusing, please clarify.

Author's response: we have clarified this sentence as follows: "High $CH_4$ concentrations in the stream suggest that external inputs of $CH_4$ — produced in the fens and transported into the stream with surface runoff, or produced in stream sediments — may have elevated emissions in Mellersta Harrsjön (Lundin et al., 2013; Paytan et al., 2015)."

L677-679: or alternatively from dilution with water with low CH4 from surface runoff and rain?

Author's response: we thank the reviewer for this suggestion.

L723: methane oxidation is also an important removal process that should contribute to imbalances between production and emission.

Author's response: One could argue that because methane oxidation rates tend to change with concentration and temperature, they would influence the flux on timescales similar to those of production (that is, timescales of a week or more). Changes in storage occur within the short residence times of CH4 gas (1-5 days). This suggest that dissociation occurs on shorter timescales – i.e. those governed by wind speed. However, in this paper we do not present a quantitative assessment of methane oxidation. Following the reviewer's earlier comment we have mentioned oxidation as a process that dissociates production from emission in the introduction.

L730-740: Wave breaking and bubbles also explain why the relation between the gas transfer velocity and wind speed is non-linear in the ocean (e.g. Wanninkhof 1992), while here you report a linear relation between gas transfer velocity and wind speed.

Author's response: the processes obtained in large water bodies are not necessarily operative in small lakes. Moreover, it is not clear from our data whether the wind-k relationship is linear or non-linear. At L. 643-645 we state "Due to the large spread of the chamber-derived gas transfer velocities (small rhombuses, Fig. 9) a power-law exponent to U10 (1.0 (0.0-1.8); exponent and 95% CI) and thus the nature of the wind-k relation could not be determined with confidence."

L763: Is thermocline tilting expected to occur in small ponds?

Yes, wind forcing can cause the thermocline to tilt in small water bodies. The extent of tilt is computed from the Wedderburn number (Imberger and Patterson, 1989). This dimensionless index takes into account stratification, wind speed, and basin dimensions. The lakes in this study are larger than ponds, albeit small, and for the relatively high winds found at these arctic sites, thermocline tilting is expected.

L797-811: Methane oxidation affects CH4 concentrations, so it's very obscure why methane oxidation should affect the alpha term. This is a scaling between gas transfer velocity that is measured and modelled, and gas transfer velocity depends on physical processes (mainly turbulence)

Author's response: in this study we infer $k_{ch}$ from measurements of the chamber concentration increase and surface concentrations. The formulations of Eq. 1 and Eq. 2 implicitly assume that all CH4 measured in the water column is emitted to the atmosphere. However, if a fraction of $CH_4$ is removed by oxidation, this would lead to an overestimation of $\Delta[CH_4]$ and an underestimation of $k_{ch}$. This in turn affects the alpha term. So oxidation does not impact gas transfer velocities directly, but may bias the gas transfer velocity high if one uses the two-layer model (Eq. 1), as is common. We have added a few sentences to the paragraph to clarify this point.

**Response to reviewer 2:**

The authors wish to thank the reviewer for their thoughtful comments and detailed suggestions, which helped improve the paper and clarify the narrative.

RC2: This manuscript documents almost a decade of weekly-monthly resolution methane concentration and flux data from 3 sub-Arctic lakes. They found Arrhenius-type temperature relationships with flux and concentration, which has been found before and suggests a strong coupling to methane production rates. They also found that wind shear drove the gas transfer velocity, but on timescales of less than a month while temperature was a driver on timescales longer than a month. They also found that stratification only played a small role in storage/accumulation and emissions in general from their systems. The methods are sound and the results are well-detailed, perhaps a bit on the long side. The dataset is quite unique as it is so long. The authors need to use the length of their dataset to substantiate their results more. They find a temperature relationship that has been shown before in quite a few other datasets, but perhaps ones not as long as theirs. Also, they find that convection does not play as large of a role in surface turbulence as has been found in other lakes. How do those datasets compare to theirs? I also strongly suggest the authors structure the discussion to highlight the main takeaway messages from this work.

Author's response: The concerns raised in the reviewer's initial statement have been addressed in our response to individual comments below.

General comments:

RC2: 1. The title seems broad as if you are referring to all lakes, but you actually point out in the manuscript many differences between your findings and those of other lakes, for example, in terms of convection contribution to k. I suggest you narrow down your title slightly. You could even highlight more in the title the amount of data that you have. This multi-year dataset is quite unique.

Author's response: We have changed the title to "Drivers of diffusive $CH_4$ emissions from shallow subarctic lakes on daily to multi-year time scales." However, we chose specify the length of the dataset in the abstract. The basic physics that control diffusion-limited emissions from water surfaces are common to all lakes though, of course, specifics such as depth and geomorphological setting will be unique to each. We feel that this is a contribution that is broadly useful.

RC2: 2. I think the discussion could do with some restructuring and more concisely define the main points of your findings. The subheadings closely follow the results structure, but this doesn't help the reader easily identify your main points. I like the way you summarized your findings in the first paragraph of the last section (summary and conclusions). I would suggest laying out the discussion with subheadings similar to the structure in that paragraph, at least to start and then edit from there. You also may not need all the information in the discussion if you find it does not highlight one of your main points.

Author's response: We believe the dataset and the variety of the analyses merits a detailed and thorough discussion. We hope the sections and subheadings as they are currently structured would allow for easy navigation to topical discussions of interest. Noting your point, we added summarizing sentences to some of the paragraphs, restructured Section 4.2 and removed section 4.7 as it does not add to the discussion. Thank you for having us revisit the organization.

Specific comments:

RC2: Line 50- should read ', of which the upper boundary..'

Author's response: We changed the sentence in accordance with the reviewer's suggestion.

RC2: Line 72 – did you not include Aben et al. 2017 because it is about ebullition? You don't specifically mention diffusive only in this sentence.

Author's response: Yes, correct. We considered the Aben et al. paper to not be directly relevant to the diffusion–limited emissions focus of our paper.

RC2: Line 101 – 'stochastic tools' sounds too vague here

Author's response: Thanks for pointing this out. We changed line 101 to "We then estimate the importance of these and other flux controls on different timescales."

RC2: Line 129 – I would say 'During the 24 hr period…' to avoid confusion. But why 2-4 samplings? What resolution and why?

Author's response: This information is detailed in section 2.8, where we write: "Chambers were sampled up to 4 times during deployment (at 10 minutes, 1–5 hours and 24 hours) which allowed us to compute fluxes at time intervals of 1 hour and 24 hours." Also the use of a short and longer time sampling provided information on those manual fluxes that might have been more episodic (i.e. affected by sub-daily changes in the gas transfer velocity) than the more regular increases that we might expect given our assumptions about diffusion-limited emissions.

RC2: Line 135-136 – you need to define Fch,unsh and Fch,sh here in this sentence (i.e., place the variables after 'shielded' and 'unshielded')

Author's response: Thanks for noting the confusion. We removed Fch,unch-Fch,sh from the equation, as we clearly state that we talk about the difference.

RC2: Section 2.2 – Do you flush the chambers between samplings or leave them the entire 24 hrs?

Author's response: This is clarified in section 2.8. We use the accumulation rate of gas in the chamber headspace to compute the flux, so we don't flush the chamber within the 24 hour deployment period.

RC2: Section 2.3 – Do you flush or mix the 4m long tube before sampling?

Author's response: Yes we do and we clarified this point in the text as follows: "the tubes were flushed by extracting a sample volume equal to the tube's volume at each location and depth."

RC2: Line 196 – do you mean 'offshore' instead of 'nearshore' here since you are differentiating between the littoral zone and another zone?

Author's response: Yes, we consider the shallow littoral zone to be near-shore, and the deeper, pelagic or profundal zone to be offshore.

RC2: Line 198 – make sure the year is correct on the reference

Author's response: Thanks, we have corrected the year.

RC2: Line 205 – define and give units for 'kch'

Author's response: $k_{ch}$ is now defined and units given.

RC2: Line 211-212 – why were there some water measurements not taken and which ones and how many?

Author's response: The design of the initial water sampling program was not intended to facilitate computation of gas transfer velocities. Simultaneous and co-located sampling was introduced in years 2016 and 2017.

RC2: Line 239 – should be 'kmod' specifically in this sentence, no?

Author's response: In our usage here, $k$ refers to the gas transfer velocity in general.

RC2: Line 245 – why do you need to do this qualitative comparison? Why is it important?

Author's response: We added a note of explanation with the following sentence: "In this way, we can assess whether the flux relations with wind speed and temperature are reproduced by the model."

RC2: Line 338 – definte 'σinit'

Author's response: This term has now been defined in the text as follows: "To allow for comparison between variables we normalized each σ-series by its initial, smallest-bin value: σnorm = σ/σinit."

RC2: Line 420 – include in the caption the panel letters for the histograms in parentheses too

Author's response: Yes that will help our explanation, panel letters have been added.

RC2: Figure 4 caption – you need to describe the squares, triangles, and diamonds in the caption itself – all the variables that you are presenting here.

Author's response: Thanks for noting our oversight, symbol descriptions have been added to the figure caption.

RC2: Figure 5 caption – what are the curves you speak of in line 500? Are you sure that e and f are the right panels when you discus the white lines on line 499? What is the resolution in panels c and d?

Author's response: Thanks for catching this; the white mixing depth lines are indeed displayed in panel f-h, not e-f. We replaced the word 'curves' with 'lines' at line 500. The resolution of the chamber flux and water concentration measurements was approximately weekly.  We hope this is evident when looking at the monthly tick mark intervals.

RC2: Table 3 title – need to describe N here

Author's response: The table title has been adjusted to reflect all variables.

RC2: Figure 6 caption – add '(a-c)' after 'residence time' and '(d-f)' after 'storage'. You mention the regressions for residence time but not for storage. Also, it looks as if there could be a trend between temperature and storage (panel e) for at least 2 of the lakes. Was there not?

Author's response: We have included the panel indicators, and fit storage quantities to Arrhenius-type exponential functions in panel e, which describe the data reasonably well ($R^2 \geq 0.70$, $p < 0.001$).

RC2: Line 560-561 – the sentence starting with 'On diel timescales..' needs rewording. I don't understand it.

Author's response: Thanks, we rewrote the sentence as follows: "On diel timescales Δ[CH4] and kmod were out of phase; Δ[CH4] peaked just before noon, when kmod reached its maximum value (Fig. 7b,d)."

RC2: Figure 7 – put a complete legend in panels a and c and state that they apply to panels b and d.

Author's response: We preferred to keep the legend as is to avoid crowding in the left panels, but we changed the symbol colour of the 1-hour fluxes to improve the clarity of the figure.

RC2: Line 612 – what is 'Twater/ice'?

Author's response: Our surface temperature sensors were frozen in the ice in winter. Because we use the whole-year temperature timeseries in our spectral analysis, we specify that this variable reflects both summer and winter variability. In the caption, we now specify "temperature of the surface water and ice".

RC2: Section 4.1 – The subheading 'Magnitude' doesn't explain much. Magnitude of what?

Author's response: We have changed the section title to 'Magnitudes of fluxes and gas transfer velocities'.

RC2: Line 632 – you obtained lower k-values by nearly a factor of 2 compared to what?

Author's response: This is in comparison to literature models. This has now been specified in the text.

RC2: Line 636 – who had the offset at 0 wind speed? You or the literature? Be specific as this sentence is a bit confusing.

Author's response: Thanks for pointing out this omission. We meant that several models in the literature have a default offset at 0 wind speed. We have amended the text as follows: "Part of the difference with the models of Vachon and Prairie (2013), Cole and Caraco (1998) and Soumis et al. (2008) was caused by the offset at 0 wind speed."

RC2: Line 637 – 'Another explanation' for what?

Author's response: Thanks for noting our oversight. This refers to the other explanation for the low k-values found in our study. We changed the sentence to specify this.

RC2: Line 639-640 – how was the atmosphere stable?

Author's response: We consider a stable atmosphere to be those periods when the tropospheric boundary layer being stably stratified, i.e. when the air temperature exceeds the surface water temperature.

RC2: Line 644-645 – I am confused because you have an equation in Figure 9 caption that has an exponent for u10 with 95% Cis.

Author's response: The equation in the caption was a linear equation, while we discussed a power-law equation in the text. We've now changed the equation in the caption to the power-law equation of Table S1.

RC2: Section 4.2 – delete 'the' in the subheading

Author's response: Thanks for noting this. 'Drivers of flux' sounds better.

RC2: Section 4.2 – this is a very important part of the discussion but I feel it needs a little more work to really bring out your main points. It reads a bit like a bunch of ideas thrown into a paragraph but without linking them all together nor highlighting why these ideas matter. For example, the first sentence states that the temperature relationship with flux and concentration suggests a strong coupling to sediment [methane] production (need that word 'methane' in there). I agree with this statement and it's an important one because you did find some nice relationships there. But the next sentence talks about stream inputs (from your own data, correct?) and then the following sentence is back to how sediment methane production could be enhanced. They seem out of order. Then the last thought about the decrease in CH4 after cold rain events is actually still in line with the temperature relationship you saw but you start this sentence off attempting to state that that shouldn't be the case if there was runoff from fens. This fens part goes more along with the streams sentence from above.

Author's response: We have revisited the organization and added two introductory sentences to the paragraph to add context to the discussion: "Methane emitted from lakes in wetland environments can be produced in situ, or be transported in from the surrounding landscape (Paytan et al., 2015). The distinction is important because some controls on terrestrial methane production, such as water table depth (Brown et al., 2014), are irrelevant in lakes.". We also replaced "cold and rainy" in the final sentence of the paragraph with "rainy", to emphasize that were are discussing horizontal transport processes here. We removed the sentence about terrestrial inputs of nutrients.

RC2: I feel the same for the second paragraph of the section. I think you clarify your point about the difference between your results and those of Read et al. I am actually not sure who had lake in the warmer, lower humidity regions – you or them? Also need to put the 50 w/m2 value in context. At the end, I wouldn't use the word 'expect' because I think you showed this. And I believe in this whole section you should already elude to the fact that these drivers work on different timescales.

Author's response:  We have rewritten this section. Read et al. (2012) did not consider Monin-Obukhov similarity scaling in their analysis.  When computing dissipation rates with it, wind shear is raised to the 3$^{rd}$ power and divided by depth whereas the contribution from buoyancy flux is only to the first power. With that constraint, buoyancy flux only drives near-surface turbulence when winds have ceased. Figure 4k shows this for our model.  Thus, differences in the meteorology between temperate and arctic lakes are not relevant here. On average, the 50 W/m2 represents the value of the net long wave radiation (Lwin - LWout) we've computed during the ice-free season in the Toolik area. We normally measured Lwin and computed LWout as a function of the surface water temperature. For reference, in our arctic work at other sites, net long wave radiation applies to periods with cloudy conditions, as often occur in the Stordalen Mire.

RC2: Line 716-728 – The first sentence of this paragraph reads more like a summary sentence. It's confusing to hear about the feedback before you describe how you got to that point. I would try restructuring this paragraph a bit. I would start with the second sentence and state it like so: 'Higher temperatures led to elevated CH4 concentrations, which in turn increased emission rates, but high wind speed was correlated with high emission rates and low concentrations. In this way,…'

Author's response: We agree and we rewrote the paragraph as suggested by the reviewer: "Higher temperatures led to elevated $CH_4$ concentrations (Fig. 4f) which in turn increased emission rates (Eq. 1, Fig. 4b) but high wind speed was correlated with high emission rates and low concentrations (Fig. 4c,g). Degassing prevented an unlimited increase of the emission rate with the gas transfer velocity. In this way, $\Delta[CH_4]$ acted as a negative feedback that maintained a quasi steady state between $CH_4$ production and removal processes throughout the ice-free season."

RC2: Line 744 – add the range of binned means in those parentheses of $0 - 10$

Author's response: The ranges have been included.

RC2: Line 784-791 – This is actually one very long sentence. Consider splitting it.

Author's response: Thank you, the sentence has been split per the reviewer's suggestion.

RC2: Line 798-799 – missing a word or something here '….but can limit surface exchange could be responsible…'

Author's response: We have split the sentence to clarify its meaning: "The observed variability in $\alpha'$ could be explained by chemical or biological factors that limit surface exchange. Such processes do not affect turbulence in the actively mixed layer, and are thus not accounted for in $k_{mod}$."

RC2: Line 834-837 – So you don't completely degas the lake, despite shallowness and frequent mixing, but you also don't have storage/accumulation of methane. I am finding a hard time reconciling those two results. I feel this needs more explanation here but also in the discussion where you mention it.

Author's response: Of course there are dynamics in the water column methane concentrations as a result of variability in the loss and input terms. Accumulation is transient – it changes on a timescale of days – and is the result of an imbalance between production and emission rates. Storage increases during long periods of stratification are not due not only due to the reduction in turbulence-driven emissions but also, in the ice-free seasons especially, to higher production rates as a result of elevated water temperatures. We rewrote section 4.3 to provide a more intuitive explanation of these processes.

[revised manuscript text omitted]
_{10}(SA) + 2.51$ | FC | 0.01–0.15 | 1–6.5 | 5.5–7.5 | Vachon and Prairie, 2013 |

[Figure]

**Supplementary Figure 1** – Based on Figure 3 (main text), but for individual lakes: Villasjön (a), Inre Harrsjön (b) and Mellersta Harrsjön (c). Comparison between gas transfer velocities from floating chambers (Eq. 2, main text) and the surface renewal model (Eq. 4, main text, with $\alpha' = 1$ and $Sc = 600$, half-hourly values averaged over each chamber deployment period). Mean ratios, and therefore $\alpha'$, are represented by the slopes of the dotted lines. Error estimates represent the 95% confidence intervals of the mean ratios.

[Figure]

**Supplementary Figure 2** – Based on Figure 7a and 7c (main text), but for individual lakes: Villasjön (**a,b**), Inre Harrsjön (**c,d**) and Mellersta Harrsjön (**e,f**). Temporal variation of the 24-hour chamber fluxes (**a,c,e**), air-water concentration difference (**b,d,f**), air and water temperature (**g**) and modelled gas transfer velocity and measured wind speed (**h**). In panels **a-f**, large squares and triangles represent binned means with 95% confidence interval error bars, horizontal bars represent binned medians and small symbols show individual measurements. Variables were binned in 10-day bins.

[Figure]

**Supplementary Figure 3** – Based on Figure 7b and 7d (main text), but for individual lakes: Villasjön (**a**,**b**), Inre Harrsjön (**c**,**d**) and Mellersta Harrsjön (**e**,**f**). Temporal variation of the 1-hour chamber fluxes (**a**,**c**,**e**), air-water concentration difference (**b**,**d**,**f**), air and water temperature (**g**) and modelled gas transfer velocity and measured wind speed (**h**). In panels **a-f**, large squares and triangles represent binned means with 95% confidence interval error bars, horizontal bars represent binned medians and small symbols show individual measurements. Variables were binned in 1-hour bins.

[Figure]

**Supplementary Figure 4** – Based on Figure 8a (main text), but for the ice-free seasons of individual measurement years. Normalized spectral densities of wind speed (**a**), air temperature (**c**), surface water temperature (**e**) and surface sediment temperature in Villasjön (**b**), Inre Harrsjön (**d**) and Mellersta Harrsjön (**f**).

**Supplementary References**

Cole, J. J. and Caraco, N. F.: Atmospheric exchange of carbon dioxide in a low-wind oligotrophic lake measured by the addition of $SF_6$, Limnol. Oceanogr., 43(4), 647–656, doi:10.4319/lo.1998.43.4.0647, 1998.

Crusius, J. and Wanninkhof, R.: Gas transfer velocities measured at low wind speed over a lake, Limnol. Oceanogr., 48(3), 1010–1017, doi:10.4319/lo.2003.48.3.1010, 2003.

MacIntyre, S., Wanninkhof, R. and Chanton, J. P.: Trace gas exchange across the air–water interface in freshwater and coastal marine environments, in Biogenic trace gases: Measuring emissions from soil and water, pp. 52–97., 1995.

Soumis, N., Canuel, R. and Lucotte, M.: Evaluation of Two Current Approaches for the Measurement of Carbon Dioxide Diffusive Fluxes from Lentic Ecosystems, Environ. Sci. Technol., 42(8), 2964–2969, doi:10.1021/es702361s, 2008.

Vachon, D. and Prairie, Y. T.: The ecosystem size and shape dependence of gas transfer velocity versus wind speed relationships in lakes, edited by R. Smith, Can. J. Fish. Aquat. Sci., 70(12), 1757–1764, doi:10.1139/cjfas-2013-0241, 2013.

---

## Author Response (AR2)

**Author's response to reviewer 1:**

We thank the reviewer for their thoughtful comments, which helped improve the manuscript.

Throughout the text specify the flux and concentration units, as grams can stand for grams of CH4 or grams of carbon.

Thanks for noting the confusion. We have now defined the units at the beginning of the paper as follows: "The flux $F$ (in mg CH$_4$ m$^{-2}$ d$^{-1}$, hereafter abbreviated mg m$^{-2}$ d$^{-1}$) depends on…"

For clarity, we have changed the units in the abstract to mg CH$_4$ m$^{-2}$ d$^{-1}$.

L 31 : Sentence "Our findings show that accurate short- and long-term projections of lake CH4 emissions can be based on distinct weather- and climate controlled drivers." is very vague and not very informative. Either remove sentence or modify it to specify which "weather- and climate controlled drivers", and also clarify what is meant by "projections". However, I'm not convinced that the authors actually demonstrate this statement by their analysis. For instance just looking at CH4 concentration, there is a general relation between CH4 and temperature in Figure 4 if all of the data is merged together, yet the CH4 concentration at a given temperature is distinctly lower in Villasjön than Mellersta Harrsjön. Yet the authors do not provide a conclusive explanation for this (depth ?). So I'm unsure if they can claim to make "accurate short- and long-term projections of lake CH4 emissions".

We agree with the reviewer that this sentence is general and vague, and have removed the sentence. The reasons for the lower concentrations of CH$_4$ relative to temperature in Villasjön than Mellersta Harrsjön are complex and related to stream inputs of CH$_4$ into Mellersta Harrsjön as well as the different exposure to wind and the stream induced difference in atmospheric stability at the two locations leading to differences in storage and flux. We discuss these explanations in Sections 3.4, 4.2 and 4.5.

L 65 : "A key control on emissions is the periodicity at which dissolved gases are brought to the air-water interface" this statement is incorrect for "dissolved gases" in general. Water bodies are not necessarily sources of gases in general and can be temporary or permanent sinks of some gases. During phytoplankton blooms water bodies can be sinks of atmospheric CO2. Conversely, in net heterotrophic systems, water bodies absorb O2 from the atmosphere. Finally, not all gases are produced in bottom waters, for instance during phytoplankton blooms O2 is produced in surface waters.

We have rewritten the start of this section as follows: "The supply of sparingly soluble trace gases to the air-water interface moderates fluxes when concentrations are higher within the water column than in the atmosphere. Trace gases such as CH$_4$ may be produced in the sediments and diffuse into the overlying water. During stratification, these gases may accumulate if the density gradient restricts the efficacy of wind mixing. Thermal convection associated with surface cooling can deepen the mixed layer and transfer stored gas to the surface, enhancing emissions (Crill et al. 1988; Eugster et al. 2003)."

L 127: explain how atmospheric pressure was regulated inside the chamber during the deployments to avoid over-pressuring (for instance at the moment of the deployment that leads to a partial compression of the gas inside the chamber) or under-pressuring when the gas was sampled (volume of gas retrieved). Over-pressure and under-pressure will artificially decrease or increase the flux measurement, respectively.

The chambers were buoyant due to the flotation devices ('pool noodles') mounted on the sides (see Figure 1 in the main text). This means that the headspace pressure inside the chamber was not substantially increased by the weight of the chamber. Moreover, the chambers were lowered into the water very carefully upon deployment, so as to minimally disturb the air and surface water, and with an open connection to the atmosphere to allow for initial pressure equilibration. The reduction in volume after three samplings (180 mL) – a fourth sampling would be prior to chamber take-out and not influence the flux – was typically less than 4%, however the minor pressure decrease this volume change may have induced would be negated by the buoyancy of the chambers.

L148: Vachon et al. (2010) also compared measurements of turbulence during deployments of chambers and on the contrary to the Ribas-Ribas work concluded that chamber deployments lead to a substantial artificial enhancement of turbulence and hence the estimate of gas transfer velocity. This seems to be an open question, and the artificial enhancement of turbulence cannot be discarded lightly. Please note that the floating chamber described by Ribas-Ribas provides gas transfer velocity values in the ocean reported by Banko-Kubis et al. (2019) that are between 2 to 10 times higher than the values predicted at the same wind speed by the conventionally accepted parameterization of Wanninkhof (2004). This would strongly suggest an artificial enhancement of gas transfer velocity measurements with floating chambers, even with the one described by Ribas-Ribas.

The measurements of Vachon et al. (2010) were performed on different chamber types (larger than our chambers, and with a square footprint). As Ribas-Ribas et al. (2018) point out, this particular chamber design may induce rolling motions that generate artificial turbulence.

Similarly to Ribas-Ribas et al. (2018), Banko-Kubis et al. (2019) performed direct, simultaneous comparisons of TKE dissipation rates inside and outside the chambers via ADV measurements. They can rule out artificial turbulence, writing: "Our data showed that the chamber did not create artificial turbulence as is assumed in Tokoro et al. (2007), as the measurements inside the floating chamber (TKEins) were not higher than outside (see Fig. 4). To the contrary, turbulence outside the chamber was 1.5 times higher than inside.". Instead, they provide several possible explanations for their high chamber k values, including fetch, surfactants, water-side convection and a shutdown of photosynthesis under the darkened chamber. Because of vast differences between model parameterizations of k (e.g. Dugan et al., 2016 and our Figure 9), a comparison between an uncalibrated model and chambers seems to be a poor indicator of artificial turbulence compared with direct ADV observations.

Gålfalk et al. (2013) also found good agreement between $k_{600}$ from a free-floating chamber similar in size and design to our chambers, and $k_{600}$ computed independently from a surface renewal model, ADV observations and an IR imaging technique.

We have carefully considered the possibility of artificial turbulence, and appreciate the pioneering work of Vachon et al. in showing that this can be a substantial problem for some chamber types. However, because the chamber design and size in Ribas-Ribas et al. (2018) and Gålfalk et al. (2013) were nearly identical to ours, and they detected no artificial turbulence across the range of wind speeds covered in our study, we think it unlikely that artificial turbulence played a significant role.

L173 : I'm aware that the FID has a linear response but it is still good practice to calibrate the FID with a standard that is relatively close to the expected values rather than using a CH4 standard of 2 ppm to measure values that are 10 to 100 times higher. This is because small uncertainty on the value of the standard, and the determination (integration) of the peak area of the standard (signal to noise ratio due to baseline fluctuations) will propagate into relatively large errors on the sample concentration computation if the difference between sample and standard values are very large, even if the response of the FID is absolutely perfectly linear.

It is vital in any trace gas measurement to accurately assess the concentration of the standard gas. Prior and after each measurement round (20-40 samples) we measured at least 10 samples of the standard gas in the same way we measured the field samples. This approach ensured a very low uncertainty (relative standard deviations of <0.25%) of the standard gas concentration.

L 854-858: I do not understand how CH4 oxidation can possibly influence the gas transfer velocity computation. You measured simultaneously a flux and a concentration from which you compute a corresponding gas transfer velocity. I do not see how methane oxidation can play a role in this computation and in the interpretation of the derived data. Sentence "This additional removal process invalidates the implicit assumption in Eq. 1 and 2 that all dissolved CH4 that we measure in the surface water is emitted to the atmosphere" does not make sense. The CH4 gradient drives the flux that both are instantaneously measured, independently of methane oxidation.

We thank the reviewer for raising this valid point, and we have amended the paragraph where we discuss the impact of biased concentration measurements on k (section 4.5):

There remains a possibility that significant methane oxidation occurs at the air-water interface, where the supply of $O_2$ meets that of $CH_4$. The actual concentration in the thin, diffusion-controlled water-side boundary layer may therefore be lower than that in the turbulence-controlled layer or in the bulk fluid, where we measure $C_{aq}$.

However, more work is needed to test the hypothesis of substantial methanotrophy at the air-water interface. We are aware of only one study that has identified methane oxidation via the bacterioneuston (Upstill-Goddard et al., 2003). Relevant statements in sections 4.5 and 5 have been nuanced to reflect this knowledge gap.

[revised manuscript text omitted]
.  More work is needed to quantify the importance of microbial removal processes at the air-water interface of freshwater ecosystems. Advanced gas transfer models can only improve the accuracy of flux estimates if they are paired with observations that capture the meteorological conditions over the lake and the spatiotemporal variability of dissolved gas concentrations. Therefore, field measurements remain necessary to inform, calibrate, and validate models. Our results indicate that the timescale of driver variability can inform the frequency of field measurements necessary to yield representative datasets for novel proxy development.

**6. Data availability**

Data are available at [www.bolin.su.se/data/](www.bolin.su.se/data/). Surface renewal model code is available by contacting SM.

**7. Author contributions**

JJ, MW and PC designed the study. Fieldwork and laboratory measurements were conducted by JJ, JS and MW. SM developed the surface renewal model code, with contributions from AC. JJ performed the analyses and prepared the manuscript with contributions from BT, PC and SM.

**8. Competing interests**

The authors declare that they have no conflicts of interest.

**9. Acknowledgements**

This work was funded by the Swedish Research Council (VR) with grants to P. Crill (#2007-4547 and #2013-5562) and by the U.S. National Science Foundation with Arctic Natural Sciences Grants #1204267 and #1737411 to S. MacIntyre. The collection of ICOS data was funded by the Swedish Research Council (#2015-06020). We thank the McGill University researchers (David Olefeldt, Silvie Harder and Nigel Roulet) for the data they provided from the carbon flux tower that was supported by the Natural Science and Engineering Research Council of Canada (#RGPIN-2017-04059). We are grateful to D. Bastviken for validating our implementation of the chamber headspace equilibration model. We thank the staff at the Abisko Scientific Research Station (ANS) for logistic and technical support. Noah Jansen created the schematic of the floating chamber pair. We thank Carmody McCalley, Christoffer Hemmingsson, Emily Pickering-Pedersen, Erik Wik, Hanna Axén, Hedvig Öste, Jacqueline Amante, Jenny Gåling, Jóhannes West, Kaitlyn Steele, Kim Jäderstrand, Lina Hansson, Lise Johnsson, Livija Ginters, Mathilda Nyzell, Niklas Rakos, Oscar Bergkvist, Robert Holden, Tyler Logan and Ulf Swendsén for their help in the field.

[revised manuscript text omitted]